# Repeated blood–brain barrier opening with a nine-emitter implantable ultrasound device in combination with carboplatin in recurrent glioblastoma: a phase I/II clinical trial

Alexandre Carpentier [1] ✉, Roger Stupp [2,3], Adam M. Sonabend [2,3], Henry Dufour[4], Olivier Chinot [4], Bertrand Mathon [1], François Ducray[5], Jacques Guyotat[5], Nathalie Baize[6], Philippe Menei[6], John de Groot[7], Jeffrey S. Weinberg [8], Benjamin P. Liu [9], Eric Guemas[10], Carole Desseaux[11], Charlotte Schmitt[11], Guillaume Bouchoux[11], Michael Canney[11] & Ahmed Idbaih[12]

Here, the results of a phase 1/2 single-arm trial (NCT03744026) assessing the safety and efficacy of blood-brain barrier (BBB) disruption with an implantable ultrasound system in recurrent glioblastoma patients receiving carboplatin are reported. A nine-emitter ultrasound implant was placed at the end of tumor resection replacing the bone flap. After surgery, activation to disrupt the BBB was performed every four weeks either before or after carboplatin infusion. The primary objective of the Phase 1 was to evaluate the safety of escalating numbers of ultrasound emitters using a standard 3 + 3 dose escalation. The primary objective of the Phase 2 was to evaluate the efficacy of BBB opening using magnetic resonance imaging (MRI). The secondary objectives included safety and clinical efficacy. Thirty-three patients received a total of 90 monthly sonications with carboplatin administration and up to nine emitters activated without observed DLT. Grade 3 procedure-related adverse events consisted of pre syncope (*n* = 3), fatigue (*n* = 1), wound infection (*n* = 2), and pain at time of device connection (*n* = 7). BBB opening endpoint was met with 90% of emitters showing BBB disruption on MRI after sonication. In the 12 patients who received carboplatin just prior to sonication, the progression-free survival was 3.1 months, the 1-year overall survival rate was 58% and median overall survival was 14.0 months from surgery.

Glioblastoma (GBM) is the most aggressive and fatal form of primary brain cancer, with an annual incidence of 3–5/100,000 people. In the United States, >13,000 patients are diagnosed with GBM each year, with a similar number in Europe. The current standard of care, established in 2005, includes maximal safe surgical resection followed by radiotherapy with concomitant and maintenance temozolomide chemotherapy and has an overall median survival of 15–20 months[1].

Despite numerous clinical trials of new therapies, the only one to show an increase in survival since 2005 is electric field therapy, which was shown to extend survival to 20.5 months in patients not progressing after chemoradiation therapy[1–4].

Treatment options at recurrence (rGBM) are even more limited, and clinical trials are typically recommended for these patients[5,6]. The few recognized treatments, such as lomustine with or without

bevacizumab have shown only minimal, if any, effects on overall survival[7]. Despite hundreds of clinical trials over the past several decades using repurposed or new compounds, no treatments have shown a significant improvement in overall survival (OS) at recurrence[8].

One of the main reasons attributed to the failure of numerous drug therapies for the treatment of primary and recurrent GBM is the presence of the blood–brain barrier (BBB), which prevents systemic agents from penetrating the brain and reaching infiltrative cancerous cells. The BBB consists of both a mechanical barrier comprising tight junctions between endothelial cells as well as active transport processes that limit the passage of drugs from the circulation to the brain[9]. A wide variety of approaches have been investigated to overcome the BBB and enhance the delivery of therapeutics to the brain, including modification of drugs or delivery routes to enhance brain exposure, as well as techniques to temporarily disrupt the BBB[10].

A promising method to enhance drug delivery to the brain is to temporarily disrupt the BBB using low-intensity pulsed ultrasound (LIPU) in combination with systemic administration of micron-sized bubbles (LIPU/MB)[11,12]. This technique is drug-agnostic, can be performed repeatedly at the time of therapy administration, and has been shown to increase the concentrations of several systemically administered drug therapies in the brain parenchyma[13–19], and to enhance survival in preclinical glioma models[20]. LIPU/MB has furthermore been shown to be safe in long-term studies after repeated BBB disruption (BBBD) in non-human primates[21–23]. This technique is now being studied in numerous clinical studies using both implantable[24] as well as transcranial ultrasound-based devices[25,26] for a variety of brain indications, including brain tumors and neurodegenerative diseases[27], and is being performed alone or in combination with small or large therapeutic agents or biologics.

A first-in-human (FIH), single-center, feasibility study of LIPU/MB was conducted with a 1 MHz single emitter implantable ultrasound device in 19 patients with recurrent GBM receiving sonications prior to carboplatin chemotherapy[24,28]. The acoustic pressure was escalated from 0.41 to 1.15 MPa, and a safe pressure level of ultrasound for repeated BBBD was determined. Treatment-related adverse events observed were transient and manageable, with no carboplatin-related neurotoxicity observed and improved outcomes for patients with clear BBBD on post-sonication MRI.

In the FIH study, carboplatin, a platinum-based drug with established anticancer effects in a variety of cancers, was used. Carboplatin has been used for over 30 years in several phase I/II clinical trials as a monotherapy in patients with high-grade gliomas but only shown a modest response[29,30]. The low activity of carboplatin against GBM can be explained by the heterogeneity of cancer cell susceptibility to the drug and by the fact that carboplatin crosses the BBB very poorly, with a brain/plasma ratio of 3–4%[31]. The only other drugs that are commonly used for treatment of GBM, lomustine, and temozolomide, have a reported brain/plasma ratio of 20–40%[31]. The concentration of carboplatin in brain parenchyma has been demonstrated to reach only 40% of the required cytotoxic dose when administered systemically at high dose levels[32]. In patients, LIPU/MB has been shown to increase the absolute brain concentration of carboplatin by 5.9-fold[33], which is consistent with the levels observed in preclinical models[34,35].

In this work, a nine-emitter implantable device, SonoCloud-9 (SC9, Carthera, Lyon, France), was developed to disrupt a much larger region of the peritumoral brain than tested in the FIH study for drug delivery and evaluated in a Phase I/II multicenter clinical trial in rGBM patients eligible for additional resection surgery and carboplatin chemotherapy. The results of this clinical trial are reported herein.

## Results

### Patient profiles

Between February 2019 and June 2021, a total of 38 patients were registered for the trial. There were four patients who signed consent and subsequently did not continue the study due to screen failures (meningitis, tumor progression, COVID-19 infection, or low platelet counts). Thirty-four[34] patients underwent tumor resection and implantation of the SonoCloud-9 device, and 33 patients underwent at least one sonication procedure associated with carboplatin chemotherapy (Fig. 1). The number of activated emitters was escalated in cohorts of 3 patients from cohort A: 3 emitters, cohort B: 6 emitters and cohort C + D: 9 emitters for ultimately a total of 27 patients treated with all 9 ultrasound emitters activated.

Details of patient characteristics are summarized in Supplementary Table 1. Patients (56% male) had a median age of 58 years, and all except two patients were treated at first recurrence (94%). Two tumors harbored an *IDH* mutation (6%). Half of the tumors were *MGMT*-methylated. Patient characteristics of cohort C (sonication first and delay to start of carboplatin) and cohort D (sonication immediately following carboplatin) were comparable except for *MGMT* promoter methylation favoring cohort C (66% methylated vs 33% in cohort D, respectively).

### Treatment characteristics

The total time for the resection surgery, including SonoCloud-9 implantation, was a mean [SD] of 3.49 [±1.95] h, while the mean estimated additional time required for the SonoCloud-9 implant placement was 24 [±13] minutes. A total of 90 sonications were performed per protocol (up to 6 cycles), with an additional 11 sonications performed off protocol (expansion for patients deemed benefitting from the treatment) in 3 patients who were not progressing after 6 cycles. Fifty-six percent (56%) of patients received ≥2 cycles of ultrasound with carboplatin. The mean overall duration of the sonication procedure was 9.9 min, which included the time to make the needle connection to the implant and the 4.5 min sonication procedure [SD: ±3.38 min]. In Cohort D ($n = 12$ patients), the mean time from the end of carboplatin infusion to sonication was 13.8 min [SD: ± 6.5] minutes, while in Cohort C ($n = 15$ patients), the mean time from the end of sonication to the start of carboplatin infusion was 63.7 min [SD:9.9], with a mean time to the end of carboplatin infusion of 120 min.

### DLT, tolerability, and safety of carboplatin delivered with concomitant LIPU/MB-based BBB opening

All patients implanted with the SonoCloud-9 were included in the safety analysis (Supplementary Table 2). The device implantation and repeated BBB opening procedure were well-tolerated in all patients. No DLTs were observed, and all nine ultrasound emitters were safely activated in all patients of cohorts C and D. Higher grade possibly or likely device-related adverse events were grade 3 pre-syncope in 2 patients, and grade 3 fatigue in one patient each. Wound infections (grade 3) following resection surgery and implantation of the device were also reported in two patients, which resolved after antibiotic treatment but led to treatment discontinuation after cycle 1 for one of the patients. Six patients (18%) complained of transient, yet severe pain (Grade 3) upon connection of the device with the needle connection procedure. No grade 4 events were recorded. One fatal event (pulmonary embolism) occurred 7 days post-surgery, but this patient did not receive any sonications/activation of the device due to the event happening prior to initiating cycle 1 and was not considered as related to the SonoCloud-9 implantation procedure as these events are frequently reported complications in GBM patients[36,37].

Lower-grade treatment-emergent adverse events (TEAEs) reported during sonication and included pain in the scalp, nausea, dizziness, headache, transient aphasia, and blurred vision, as shown in Supplementary Table 2. These presumed focal deficit TEAEs resolved within

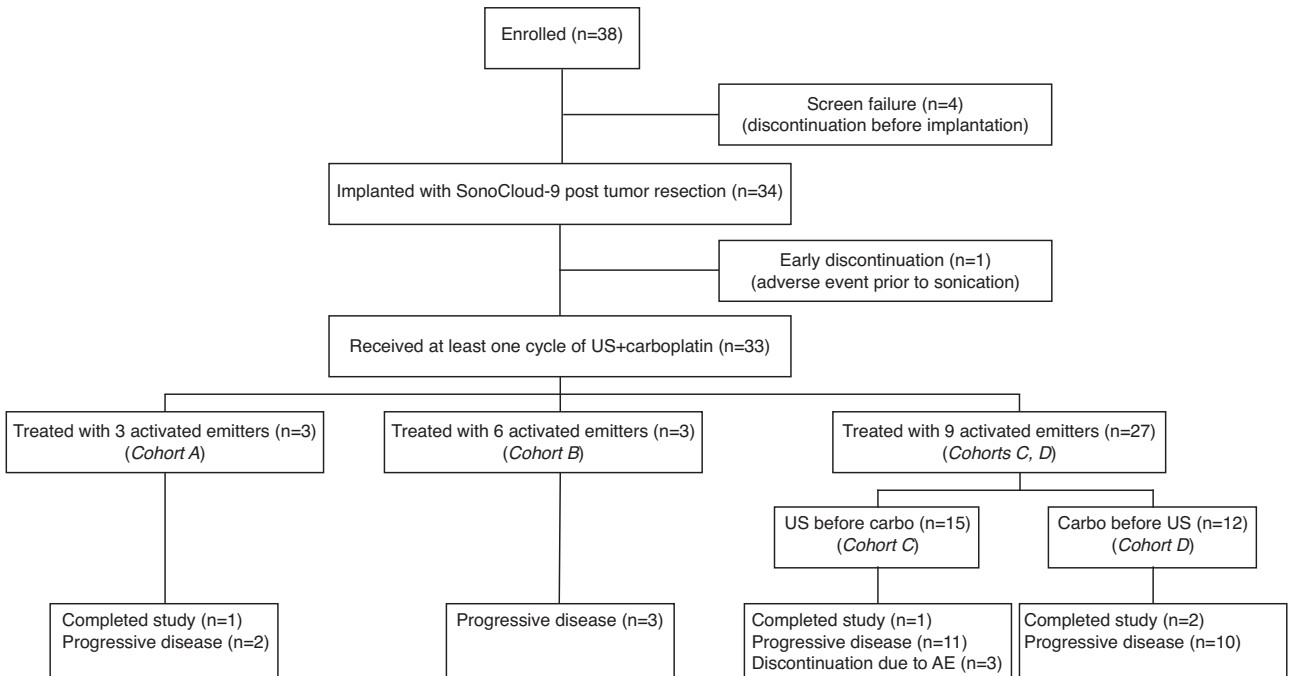

**Fig. 1 | Trial flow diagram.** A total of 38 patients were enrolled in the trial, with 33 patients being implanted and receiving at least one sonication with the SonoCloud-9 device to disrupt the BBB. One patient had early discontinuation after implantation due to a pulmonary embolism before any device activation occurred. A total of 27 patients were treated with all 9 emitters of the device, with 15 patients treated with ultrasound (US) before carboplatin administration and 12 patients treated US immediately after carboplatin administration. A total of 4/33 patients completed the study and received six cycles of BBB disruption with the SonoCloud-9 at the time of carboplatin administration.

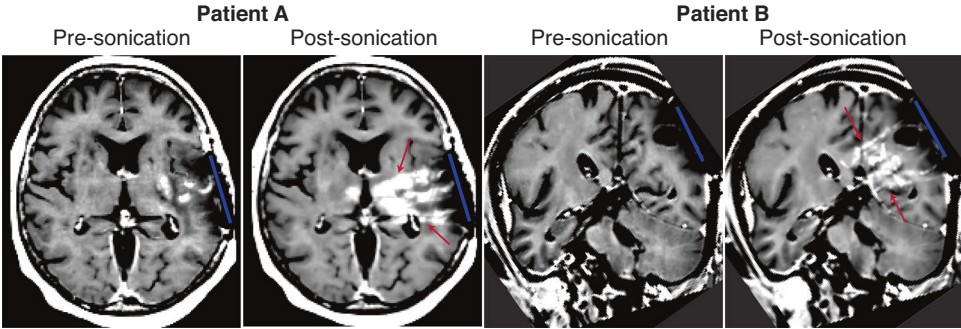

**Fig. 2 | BBB disruption case studies.** Pre and post-sonication images from two patients with nine emitters active showing the region of BBB disruption induced by the SonoCloud-9 System (red arrows indicated region of BBB disruption; blue line indicates position of SonoCloud-9 device).

15 min after the sonication procedure, except for one patient who reported blurred vision (Grade 1) that lasted for three months. None of these low-grade TEAEs prevented patients from receiving additional cycles of sonication and carboplatin administration.

The pattern of hematological and non-hematological toxicities secondary to carboplatin (e.g., bone marrow suppression, anemia, and vomiting) was as expected. Neutropenia led to discontinuation of treatment after cycle 1 for two patients (6%) not recovering within the delay specified between cycles in the protocol.

A total of 52 pre and post ultrasound T2*/SWI pairs were reviewed by a trained neuroradiologist (BPL), corresponding to 52 sonications in 33 patients. In 36/52 of the images (69%), at least one or more pre-existing hypointense regions were present in the sonication field. In all cases, these pre-existing hypointense SWI signal abnormalities remained similar in the post-US image. A single new hypointense SWI signal abnormality (<5 mm in diameter) was observed in the sonication field in post-US SWI images for 6/52 treatments (11%). In all cases where a later follow-up SWI image was available (i.e., following months), no hypointense signal increase was observed.

### SonoCloud-9 BBBD

The extent of BBBD was visualized by gadolinium-uptake on T1w MRI performed 24–48 h before and immediately (within 60 min) after sonication (cycle 1 to cycle 3, when performed). Representative images for two different patients are shown in Fig. 2. BBB opening was further evaluated using 61 LIPU/MB procedures in 27 patients in which all nine emitters were activated (Cohorts C, D). Of these 27 patients, we performed additional imaging after cycles 2 and/or 3 providing for a total of 34 additional post-sonication images. A median depth of BBBD of 64 mm was observed on T1w contrast-enhanced imaging, and 90% (95% CI: [0.63; 0.98]) of activated emitters led to grade 2–3 BBBD, using the grading criteria previously described[28,38]. No difference between the first and subsequent sonications was observed. The absence of clear BBBD (grades of 0 or 1) was explained by the limitation of the automatic method to analyze poor-quality images, and

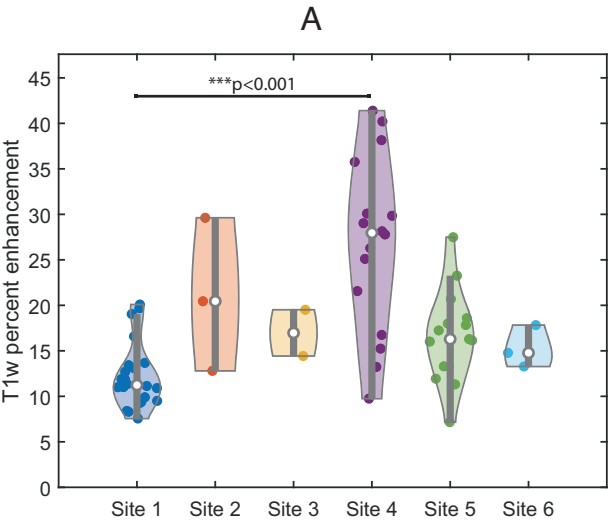

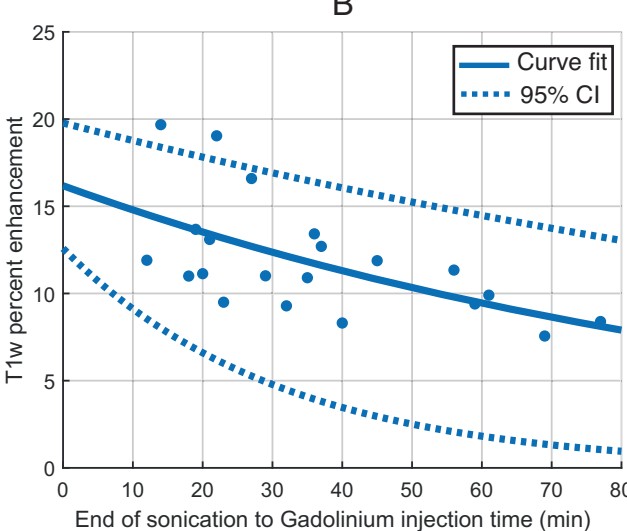

**Fig. 3 | BBB disruption by site and closure dynamics.** Post-sonication MRI was performed for all patients enrolled on trial after the first three sonication sessions to verify BBB opening. **A** A significant difference in sonication-induced T1 contrast enhancement was found between sites due to MRI acquisition parameters and gadolinium contrast agent used, with Site 4 using Gadavist® and all other sites using Dotarem®. $N = 61$ sonications (in 27 different patients from cohorts C and D). Site 1: $N = 22$; Site 2: $N = 3$; Site 3: $N = 2$; Site 4: $N = 16$; Site 5: $N = 15$; Site 6: $N = 3$. The violin plots indicate the median (with dot), first and third quartiles (gray line), and min and max (colored contour). ANOVA testing was performed with post hoc

Tukey–kramer (site 1 and 4: $p = 0.0004$). **B** The time between sonication and gadolinium bolus for T1w image acquisition was 10–77 min at Site 1, which was due to the availability of MRI after sonication. There was a significant negative correlation between enhancement intensity and sonication to gadolinium injection time for treatments performed at this site (in 10 different patients from cohort C and D treated in site 1, $p = 0.05$), with an exponential decay fit indicating a half-closure time for the BBB of 1.3 h. The dashed line indicates the 95% confidence interval (CI) for the exponential decay fit. Source data are provided as a Source Data file.

**Table 1 | Clinical outcomes of patients with nine emitters activated (cohorts C and D)**

| Cohort | mPFS (months) [95% CI] | 9-month OS [95% CI] | 1-yr OS [95% CI] | mOS (months) [95% CI] |
|---|---|---|---|---|
| C + D ($N = 27$) | 2.6 [2.2, 3.3] | 70% [0.49, 0.84] | 52% [0.32, 0.69] | 12.0 [8.4, 14.0] |
| C carboplatin after US ($N = 15$) | 2.5 [2.1, 2.8] | 67% [0.38, 0.85] | 47% [0.21, 0.69] | 11.8 [8.0, 13.2] |
| D carboplatin before US ($N = 12$) | 3.1 [2.1, 5.0] | 75% [0.41, 0.91] | 58% [0.27, 0.80] | 14.0 [6.7, 17.3] |

Patients receiving carboplatin before ultrasound (US) in Cohort D had a median OS of 14.0 months, in comparison to patients who received carboplatin after the USA, who had a median OS of 11.8 months. The median and 95% confidence interval (CI) are indicated for the progression-free survival (PFS) and overall survival (OS).

regions with a large volume of resection cavity or tumor enhancement that confounded the algorithm.

Significant differences in the extent of BBBD were observed between participating investigational sites, while results across patients treated at the same center were comparable. The semi-quantitative method to evaluate BBBD intensity based on post-sonication MRI is dependent on acquisition parameters: sequence, scanner, and contrast agent used. As shown in Fig. 3A, the enhancement intensity was significantly higher at site 004 where Gadavist® was used, than in site 001 ($p < 0.001$), where Dotarem® was used. This greater enhancement intensity in the brain of some contrast agents has been reported previously[39].

Depending on the availability of the MRI after sonication, the time between sonication and administration of the gadolinium bolus varied from 10–77 min range (mean: $33 \pm 14$ min) depending on the immediate availability of the MRI scanner. As shown in Fig. 3B, a significant negative correlation was found between enhancement intensity and sonication to gadolinium time for treatments performed with sufficient data (site 001, analysis on 31 sonications, Spearman correlation, rho = $-0.6$, $p = 0.005$). An exponential decay fit indicated a half-closure time of LIPU-disrupted BBB of 1.3 h (95% confidence interval: 0.4–2.2 h).

### Clinical outcomes
A summary of the clinical outcomes is shown in Table 1. The median overall survival of patients treated with all nine emitters activated was

12.0 months (95% CI (8.4, 14.0). The PFS could be considered equivalent in both cohorts (3.1 months [95% CI: 2.1–5.0] *vs* (2.5 months, 95% CI: 2.1–2.8], $p = 0.55$). The 1-year OS rate was 58% [95% CI: 0.27, 0.80] in Cohort D and 47% [95% CI: 0.2, 0.7] in Cohort C. The median OS was 14.0 months [95% CI: 6.7, 17.3] in Cohort D and 11.8 months [95% CI: 8.0, 13.2] in Cohort C.

### Comparison of the sequence of administration of carboplatin
A post hoc analysis was performed to evaluate the effect of the treatment sequence by comparing tumor growth and overall survival in patients treated with all nine emitters in Cohorts C and D. Cohort D evaluated a change in treatment sequence, i.e., sonication to follow immediately after the end of carboplatin infusion and thus also shortening of the interval between chemotherapy administration and sonication to a few minutes only (mean interval between sonication and chemotherapy 64 min versus 14 min in cohorts C and D, respectively). Better in-field tumor control was observed in patients in cohort D compared to cohort C (chemotherapy administration up to 60 min after sonication). An example of the T1w contrast-enhanced evolution of the tumor volume from 6 monthly pre-sonication images is shown in Fig. 4. This patient had an increase in T1 enhancement due to tumor up to Cycle 2 that decreased over time during the monthly treatments.

The evolution of the tumor-related hyperintense T1w volume in the region targeted by the implant (shown in green in Fig. 4) was evaluated and is shown in Fig. 5A for 26/27 patients treated with nine

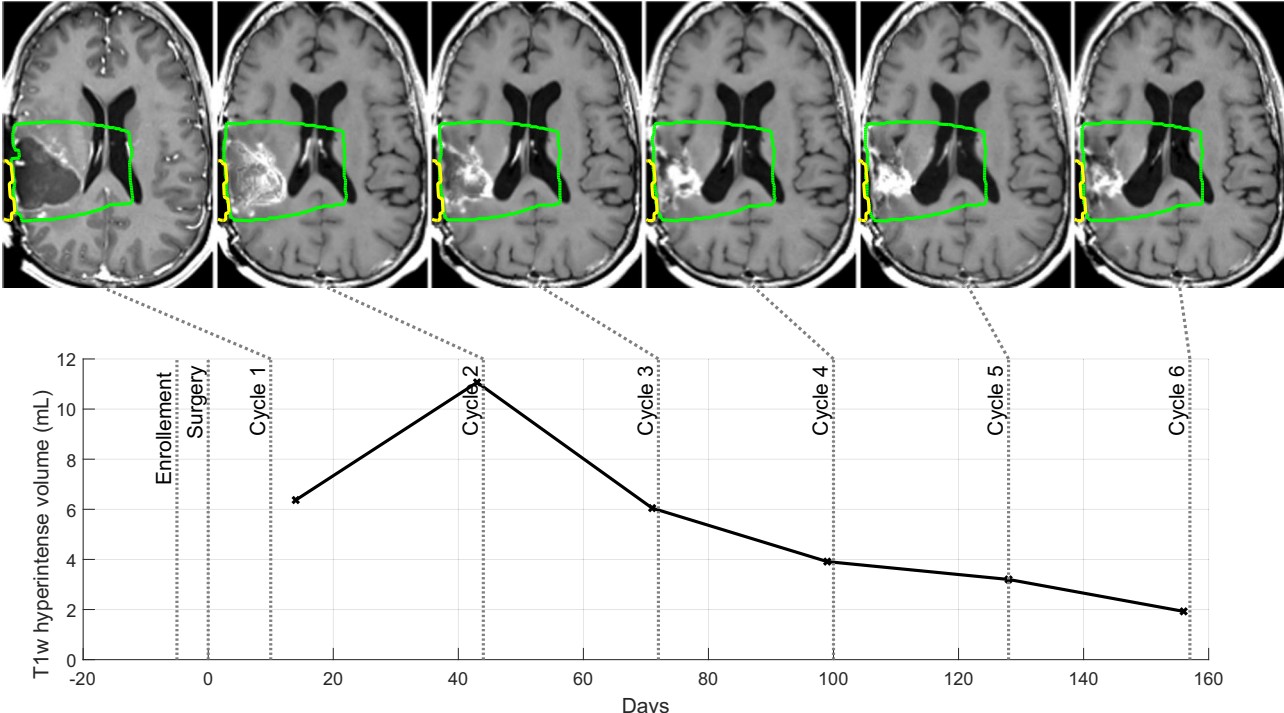

**Fig. 4 | Patient case study of radiological response.** Overall, tumor growth was better controlled within the field of sonication compared to outside the field of sonication in patients treated in Cohort D that received carboplatin infusion prior to sonication to disrupt the BBB. An example of the T1w contrast-enhanced evolution of the tumor volume from six monthly pre-sonication images is shown. This patient had an increase in T1 enhancement up to Cycle 2 that then decreased over time with each monthly cycle of treatment. The region outlined in green corresponds to nine cylinders, each 20 mm × 80 mm in front of each of the emitters of the ultrasound implant, which corresponds to the sonicated volume with an additional diffusion margin of 5 mm. Source data are provided as a Source Data file.

emitters in Cohorts C and D. One patient in Cohort C was excluded as they left the study after cycle 1 due to an adverse event (not tumor progression) and additional MRIs beyond pre-cycle 1 were not available. As shown in Fig. 5B, a significantly lower tumor growth rate over the study duration was found in cohort D (median = 0.54 mL/month), than in cohort C (median = 2.31 mL/month) (Wilcoxon–Mann–Whitney test: $p = 0.04$). When the region targeted by the implant was excluded from the analysis, and only regions outside the sonication field were used, there was no significant difference between the evolution of the T1w enhancement between cohorts C and D ($p = 0.55$).

In order to further evaluate the effect of the treatment sequence, an analysis was performed using the end-of-study MR images from the patients in Cohorts C and D. The contrast-enhancing tumor at the end-of-study was segmented with a semi-automatic method in the post-gadolinium T1w images. The percentages of brain volume comprised between two given distances to emitters axes (tube-shaped regions) covered by tumor mask were computed. An analysis from a representative patient of this analysis is shown in Fig. 5C, in which the nine emitters and distances from each emitter axis are shown along with the region of enhancing tumor volume from the end of study images (red contour). Figure 5D shows a comparison of the local tumor progression probability metric between patients from Cohorts C and D using this analysis method. There was less likelihood for there to be tumor in the cylindrical zone in front of each emitter for cohort D than for cohort C, up to a cylinder with a radius of 10 mm (with statistical significance up to 7.5 mm, Wilcoxon rank-sum test). Further away from the emitters, no difference between the two cohorts was observed.

## Discussion

The field of BBBD by ultrasound is rapidly growing, with dozens of clinical trials on-going[40]. The technique can be used for a wide range of brain diseases and in combination with multiple drug therapies. Currently, the SonoCloud-9 is in clinical trials with temozolomide for newly diagnosed GBM patients (NCT04614493), nab-paclitaxel (Abraxane®), and a combination of nab-paclitaxel/carboplatin in rGBM patients (NCT04528680). The SonoCloud-1 is being investigated in combination with checkpoint inhibitors in brain metastases (NCT04021420), and results from a Phase 1 clinical trial were recently reported using the SonoCloud-1 in Alzheimer's patients[41]. Furthermore, the technique of using ultrasound for temporarily disrupting the BBB is being explored as a tool to enhance liquid biopsy for diagnostic purposes[42].

In our FIH study, we demonstrated that it was safe to repeatedly disrupt the BBB in patients with rGBM every month in combination with carboplatin chemotherapy using a 1 MHz single emitter, 10-mm diameter implantable ultrasound device designed to fit in a burr hole in the scalp[28,43]. Here, we showed that the repeated activation of a nine-emitter implantable ultrasound device (SonoCloud-9), developed to replace a bone flap and to disrupt the BBB over a volume nine times larger than our initial FIH study, is safe. No DLTs were observed, and repeated procedures every four weeks were well-tolerated in patients receiving a target dose of AUC5 of carboplatin. Although 6/34 patients reported severe pain, these events were due to the needle puncture at the time of sonication with a duration of several minutes did not prevent the patient from receiving treatments, and were not attributed to the ultrasound emission itself. Additional steps, including the use of analgesic creams prior to the needle connection procedure have been incorporated into on-going and future clinical trials with the Sono-Cloud device to minimize this patient discomfort.

The extent of BBB opening in this study encompassed an area of peritumoral brain tissue up to >6 cm deep from the brain surface and was shown by transient gadolinium-uptake post sonication. The implant has flexible emitters to conform with the curvature of the skull. The nine emitters of the implant are activated sequentially, such that there is no direct acoustic interference from adjacent emitters. Nevertheless, there was some overlap at about 5 cm from the inner

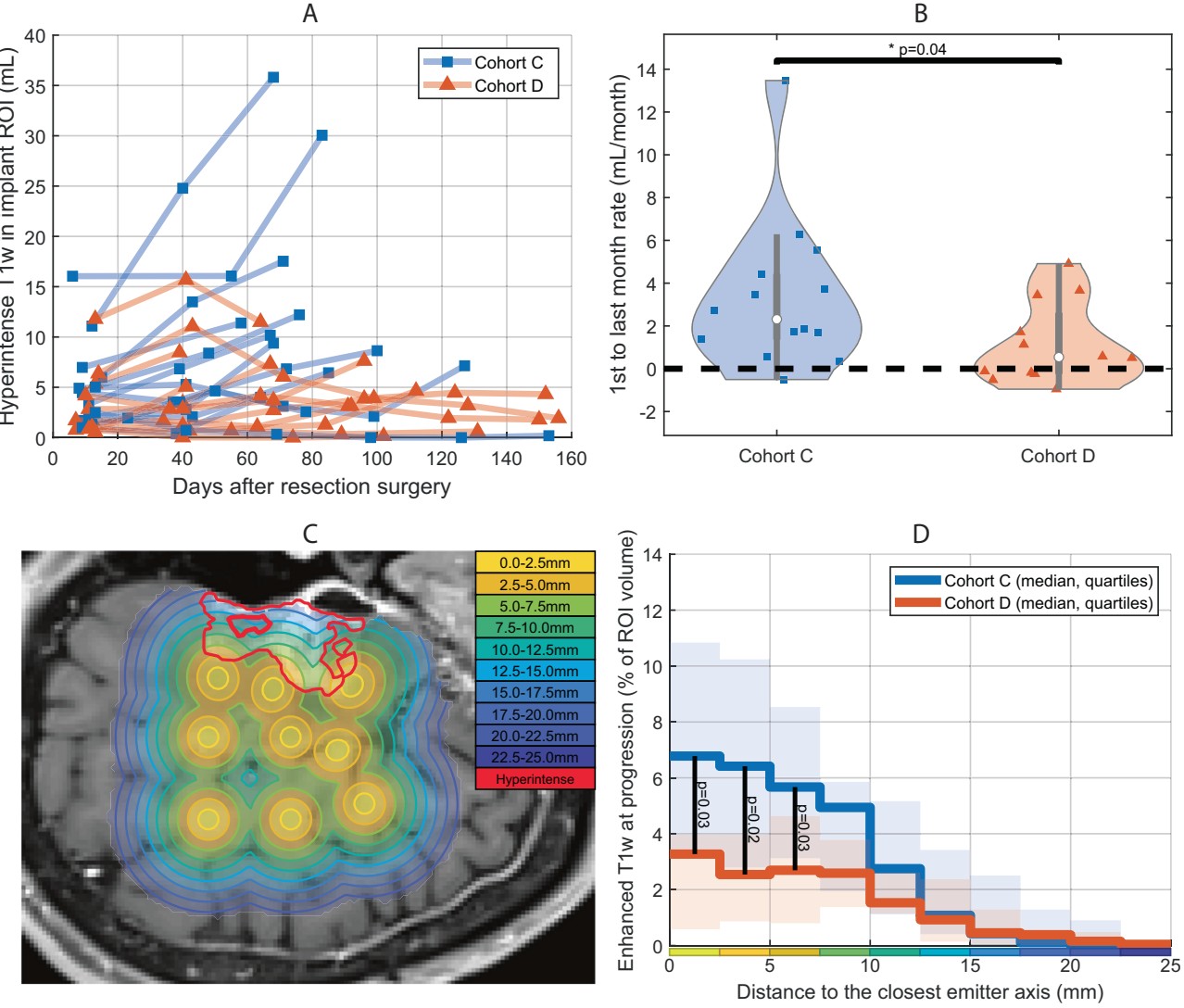

**Fig. 5 | Tumor growth was slower in patients receiving carboplatin before sonication.** The evolution of the tumor-related hyperintense T1w volume in the region targeted by the implant (shown in green in Fig. 4) was evaluated and is shown in **A** for both cohorts treated with 9 emitters active. A significantly higher growth rate over the study duration was found in cohort C (median = 2.31 mL/month) than in cohort D (median = 0.54 mL/month), as shown in **B** (Two-sided Wilcoxon–Mann–Whitney test: $p = 0.04$). The violin plots indicate the median (withe dot), first and third quartiles (gray line), and min and max (colored contour). When the region targeted by the implant was excluded from the analysis, there was no significant difference between the evolution of the T1 enhancement ($p = 0.55$). The local probability of tumor control was evaluated using T1w images at progression. A visualization of this analysis is shown in **C** in which each of the circles depicts an emitter axis from a real SonoCloud-9 implant in a patient. **D** The percentage of ring-shaped ROIs surrounding emitter axes covered with hyperintense tumor at the end of the study were compared between Cohorts C and D. The probability of T1w enhancement was lower in Cohort D than in Cohort C (two-sided Mann–Whitney $U$ test, $N = 26$, $p < 0.05$ for radiuses up to 7.5 mm). The sonicated zone with BBB disruption corresponds to the 0–5 mm bin (10-mm cylinders), and effect on local tumor progression is observed up to 10-mm from the emitter axes (statistically significant up to 7.5 mm). Source data are provided as a Source Data file.

skull/emitter surface in tissue that may have received sonications from multiple beams. In this study, the brain volume with two beam trajectories or more overlapping represented an average of <10% of the sonicated volume (9.1 +/-3.9%). The simulated acoustic energy in this overlap volume was always lower than the maximum in the targeted tissues (−60% +/−20% on average), because of diffraction and attenuation before the beam trajectories crossed. No differences were observed between T1w enhancement from BBBD when comparing zones with and without overlap.

The intensity of the gadolinium signal in the brain tissue was a function of the delay between sonication and gadolinium injection/image acquisition. Interestingly, we observed differences in the extent of gadolinium uptake between research institutions that used different brands of gadolinium-based contrast agents. Our findings suggest that

the duration of BBB opening may be shorter (measured here to be 1.3 h) than previously reported in the preclinical literature (2–24 h) and that the continuous process of restoration of the BBB in the non-tumoral brain tissue begins immediately after the end of sonication[16]. These results are in agreement with those recently published in a separate patient cohort treated with the SonoCloud-9 in which there was rapid restoration of the BBB within 1 h of sonication[33].

New small (<5 mm) hypointense regions were observed in 6/52 treatments when comparing available pre and post-sonication T2*/SWI image pairs. In routine clinical practice, none of these would have been considered to be clinically significant due to their small size and lack of mass effect according to a trained neuroradiologist (BPL) in the setting of postsurgical MRI follow-up after neurosurgical tumor resection or after any type of chemotherapy or radiation therapy. In a recent study

using a transcranial ultrasound system to disrupt the BBB, new T2* changes were observed in 50% of the targeted regions after BBBD, but no evidence of tissue damage was observed following surgical removal of this tissue 2 to 4 h following treatment. The authors furthermore proposed such changes may be due to local gadolinium or protein extravasation[44]. These SWI features are also frequently observed in GBM patients due to radiosurgery or tumor angiogenesis and have been proposed as a way to distinguish high-grade gliomas from low-grade ones[45].

Given the short BBB half-life observed in patients, the timing of drug administration vs. LIPU/MB is likely a key parameter for optimizing brain drug concentrations and the efficacy associated with the drug. In cohort D, carboplatin administration (an i.v. infusion over 30–60 min) immediately preceded sonication, while in the initial cohorts, carboplatin chemotherapy was delivered after a delay of up to 90 min after sonication. In cohort D, sonication thus occurred when plasma concentrations of carboplatin were at their peak. Patients treated in cohort C and D, which differed by the sequence and delay of chemotherapy administration, suggest that improved outcomes in cohort D may indeed be clinically relevant, and this sequence of sonication/drug administration will be further investigated in future clinical trials.

Preclinical studies, all performed with drugs administered at the time of sonication (and typically with a much shorter infusion duration than in humans), demonstrated a 4–7 fold enhancement in carboplatin levels after sonication to disrupt the BBB[34,35]. As part of the trial described herein, a subset of patients participated in an exploratory ancillary study in which carboplatin was administered IV in the operating room over 15 min (test dose, AUC 3.5) immediately after intraoperative LIPU/MB of non-enhancing peritumoral tissue and subsequent biopsy and resection of the brain tissue. These studies demonstrated that carboplatin levels were enhanced by 5.9-fold in the sonicated peritumoral brain compared to non-sonicated regions, confirming brain ratio observed in preclinical studies[33]. A closer time of administration of carboplatin to sonications may therefore lead to a higher concentration of the drug into the brain resulting in a lower tumor growth rate, as observed in cohort D.

This apparent better control of tumor growth did not translate, however into a clear improvement of the PFS, as evaluated by the RANO criteria. There tended to be a transient increase in T1w contrast enhancement up to the 2nd month of treatment that then decreased over time (Fig. 5). In GBM, radiological evidence of pseudoprogression, involving increases in enhancement T1 images that are not necessarily due to tumor progression, is a well-known phenomenon after upfront treatment with radiotherapy and temozolomide and this non-specific contrast enhancement that can result is a potential limitation of the tumor growth analysis performed in this study[46,47]. In our study, MRIs were performed every month before each cycle of chemotherapy, which may have led to premature report of radiological progression (and thus treatment discontinuation) in comparison with standard imaging follow-up (typically every 8–12 weeks), and therefore underestimated PFS. In future trials, confirmatory, less frequent MRIs or the allowance of treatment continuation if there is only evidence of radiological and not clinical progression may be advised, and central review according to updated RANO criteria will be used to provide a more accurate evaluation of time to progression.

The mOS of 14.0 months (95% CI (6.70, 17.3) obtained with the optimized sequence of carboplatin treatment in Cohort D was encouraging, when considering that two-thirds of the tumors were MGMT unmethylated and all were IDH wild-type[48,49]. For the patient population treated in this trial, the expected mOS is between 8–12 months, with patients eligible for additional resection surgery typically having longer survival. Interpretation of these results is limited due to the small cohorts (15 patients in Cohort C, 12 patients in Cohort D), the fact that Cohorts C and D were not recruited in parallel,

and considering that only six clinical sites participated. Nevertheless, these findings support the hypothesis that the improved efficacy is consistent with higher concentrations of drugs in the brain at the time of sonication.

Our approach has shown the potential to temporarily disrupt the blood–brain barrier using a nine-emitter implantable ultrasound system, thus offering new opportunities for enhancing drug delivery and treating brain diseases with high unmet needs, such as GBM. Its clinical efficacy is being further evaluated in a larger pivotal trial (NCT05902169).

## Methods

### Study design

This study was a prospective, open-label, international, multicenter, single-arm, dose-escalation, phase I/II clinical trial enrolling rGBM patients. This trial was sponsored by Carthera and was conducted in accordance with the criteria set by the Declaration of Helsinki. The study was performed at four clinical sites in France and two sites in the USA. All patients provided written informed consent in accordance with institutional guidelines. Approval was obtained from the ANSM (French National Health Agency), the FDA (Food and Drug Administration), local institutional review boards (IRB), and the Sud Méditerranée V ethics committee (CPP). The study was conducted in accordance with good clinical practices. The trial was registered as NCT03744026, EudraCT 2014-000393-19, and BRC: 2018-A01511-54. The study began in February 2019, and follow-up was completed in November 2022.

### Patient selection

Patients experiencing recurrence (any) of a histologically proven primary GBM, after at least a first-line standard of care (radiation with concurrent and adjuvant temozolomide) were recruited. Qualifying patients with good performance status (KPS ≥ 70), were eligible for carboplatin-based chemotherapy and tumor resection, with tumor size limited to 70 mm in diameter on T1w contrast-enhanced MRI. Patients receiving steroids, should be stable and have received <40 mg prednisone dose (dexamethasone ≤6 mg) for at least 7 days preceding study participation.

### Trial design

This trial was designed to evaluate the safety of concomitant carboplatin administration with transient disruption of the blood–brain barrier by LIPU using the SonoCloud-9 implantable device as well as the performance of the SonoCloud-9 device to repeatedly disrupt the BBB. The trial first evaluated the dose-limiting toxicity (DLT) of escalating numbers of ultrasound beams (3, 6, and 9 beams) at constant acoustic pressure (1.03 MPa) using a 3 + 3 dose escalation design (cohorts A, B, and C) and then confirmed the safety and efficacy of BBB opening in two expansion cohorts (cohorts C & D).

For Phase 1 (escalation of the number of active emitters/beams), the primary objective was to identify the maximum tolerated dose (MTD) defined as the highest active beam level at which ≤1 DLT occurred in a maximum of 6 patients by cohort. The DLT was defined as any Common Terminology Criteria for Adverse Events (CTCAE), version 5.0, Grade 3 or higher, event at least possibly attributable to the sonication or to the sonication plus carboplatin procedure that occurred within 15 days and that did not respond to optimal medical management (including steroids) within 7 days, including symptomatic intracranial hemorrhage of any grade, seizure of grade 3 or 4 (status epilepticus) regardless of time of resolution or symptomatic stroke of any grade. The MTD was defined as the highest active beams level at which ≤1 DLT occurs in a maximum of 6 patients by cohort. In Cohort A, SC9 at the 3 active beams level was given to 3 patients. In Cohort B, SC9 at the 6 active beams level was given to 3 new patients. In the absence of DLT, SC9 at the 9 active beams level was given to 3

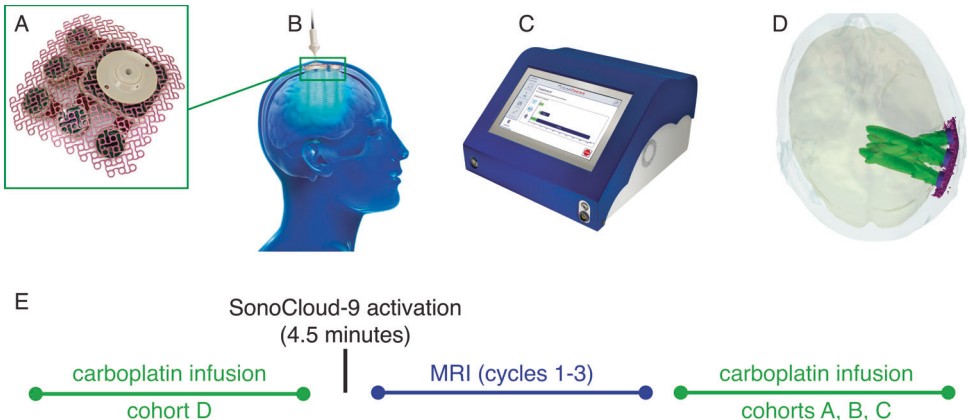

**Fig. 6 | SonoCloud-9 system.** The SonoCloud-9 System consists of three components: **A** an implant containing nine 1-MHz, 10-mm diameter ultrasound emitters that are powered by **B** a transdermal needle used to connect the device at each activation to **C** an external generator that includes a touchscreen interface to guide the treatment and provide the energy to the implant. At each activation of the device, the nine emitters of the implant are activated sequentially using 25-ms long pulses every 2 s (duty cycle = 1.2%) at the same time as an intravenous infusion of ultrasound resonators (Definity®, 10 µL/kg) for a total duration of 270 s. **D** The simulated region of BBB disruption (corresponding to a region of pressure >0.2 Mpa) in a hypothetical patient. **E** The activation procedure to disrupt the BBB was performed monthly at time of carboplatin infusion, with carboplatin infusion performed either immediately before (cohort D) or after (cohorts A, B, C) sonication. **A**, **C** reproduced with permission from Sonabend et al.[33].

new patients (cohort C). If 0 of the 3 patients experienced a DLT, then the 9 active beams level was determined to be the MTD, and those patients of groups C of the phase 1 study were added to the expansion phase 2a patients treated at MTD (groups C, and D), to assess the BBB opening at the maximum number of SonoCloud-9 emitters tolerated. An Independent Data Safety Monitoring Board evaluated safety data and advised the continuation of the trial after each cohort of the dose escalation portion.

In the extension Phase 2a, the primary objective was to evaluate the BBB opening efficacy, which was evaluated as the percentage of successful ultrasound sessions. A successful ultrasound session was defined by the number of emitters for which the BBB opening was Grade 2 (subarachnoid and gray matter contrast enhancement) or Grade 3 (subarachnoid, gray & white matter contrast enhancement) during the first three cycles by comparison of pre- and post LIPU session T1w magnetic resonance imaging (MRI) as defined by in our previous work[28,38].

The secondary endpoints included the frequency and severity of adverse events (incidence of adverse events summarized by system organ class and/or preferred term and severity) based on the CTCAE, version 5.0, the time to and localization of recurrence(s) on magnetic resonance imaging, feasibility of the procedure considering the time required for the sonication (from the beginning of needle connection to the end of ultrasound emission and needle extraction) at the first 3 sonications of each patient, and the total time for SonoCloud-9 positioning using the SonoCloud 9 template. Six-month progression-free survival (6m-PFS), median progression-free survival (mPFS), and 1-year overall survival (1y-OS) and median OS (mOS) were also evaluated. In this trial, time to progression or to death was calculated from the time of surgery/device implantation to the time of events.

For each patient, participation was planned to last ~7.5 months: from inclusion to surgery was a maximum of 14 days, from surgery to first sonication (Cycle 1) was a minimum of 9 days to a maximum of 14 days to allow for surgery recovery. Patients were then treated until month 6, tumor progression, or premature discontinuation, whichever came first. The number of cycles was defined by the chemotherapy frequency (every 4 weeks). The outcome was documented at the end of the study visit that took place within 1 month of the event. Patients who did not progress by month 6 were allowed to continue ultrasound treatment (sonication with chemotherapy) as initiated is considered to be in the best interests of the patient by the Investigator.

### Patient eligibility
Patients eligible for tumor resection surgery and carboplatin chemotherapy with histologically proven recurrent de novo GBM after at least a first-line standard of care (maximal safe resection, if feasible, radiation with temozolomide, then maintenance temozolomide) were proposed to participate in the trial. The pre-surgery tumor size was limited to a maximum diameter of 70 mm T1w contrast-enhanced MRI. Patients with multi-focal and posterior fossa tumors were not eligible. Patients with KPS < 70, at risk of surgery site infection, and patients who had undergone antiangiogenic treatment or patients in need of continuous antiplatelet therapy were also excluded. The dose of steroids was limited to 40 mg of prednisone (or dexamethasone ≤ 6 mg) for at least 7 days, at inclusion. The use of non-absorbable hemostatic agents or dura matter substitutes was not authorized at surgery.

### Device implantation
The implantation of the SonoCloud-9 device (Carthera, Lyon, France), shown in Fig. 6, was performed by a trained neurosurgeon at the end of planned standard tumor resection. The SC9 implant is designed to replace a 58 mm × 58 mm bone flap that is removed during surgical resection. At the beginning of the surgery, after skin opening, the surgeons positioned a template using a neuro-navigation pointer to ensure that the implant would cover the maximum infiltrative region surrounding the tumor resection bed (high-signal FLAIR region). Once the SC9 implant location was set, the template was used to trace the craniectomy size on the skull of the patient for the implant location. The surgeons performed a standard craniotomy, opened the dura mater, and performed the tumor debulking/resection as per the routine. Then the dura matter was closed, and the SC9 implant was then secured on a window in the skull epidurally and recovered by the skin.

### Sonication procedure
The SC9 device contains no internal energy source and is activated on demand by connecting the device to an external generator using a transdermal needle, as shown in Fig. 6. Each sonication step consists of the generation of sequential pulses from each emitter (1 MHz, 25 ms pulse, 0.5 Hz, 270 s) in combination with the IV administration of the ultrasound resonator (microbubbles [MB], Definity®/Luminity® 10 µL/kg, Lantheus, N. Billerica, MA). The peak pressure amplitude generated

by each ultrasound emitter of the SonoCloud-9 device, which was pre-calibrated in water before implantation, was set to 1.03 MPa. A unique calibration coefficient for each emitter was read by the SonoCloud generator to set the electrical power sent at the activation of each emitter. The ultrasound resonator was injected as a 30-s bolus at the start of the ultrasound sonication procedure. Sonication was performed prior (within 60 min) to the carboplatin chemotherapy at cycles 1, 2, and 3 and no more than 30 min before the start of carboplatin therapy for the following cycles, for cohorts A, B & C. For cohort D, the sonication was performed immediately after the completion of the carboplatin infusion. An illustration of the sequence of carboplatin infusion/device activation is shown in Fig. 6E. After completion of the six sonication cycles specified in the protocol or in the case of progression/premature discontinuation of the trial and prior to the end of the study visit, the removal of the device could be performed unless considered as contra-indicated by the investigator or refused by the patient. Timing and duration of implantation and sonication procedures were collected on the eCRF from sites participating in the trial.

## Carboplatin administration

Standard carboplatin chemotherapy using any FDA/ANSM-approved, therapeutically equivalent carboplatin injectable drug product, with a target AUC of 4–6 mg/ml*min (AUC as per local practice and investigators discretion) was given according to the Calvert formula[50]. Cycles were to be repeated every 4 weeks provided the absolute neutrophil count had recovered to ≥1500 cells/mm$^3$ and the platelet count was at least 100,000 cells/mm$^3$. Subsequent dosages were adjusted for toxicity as needed per local practice.

## Study assessment

All patients were clinically assessed at least once a month (prior to the next cycle) and included standard laboratory blood analyses (complete blood counts, chemistry with liver and kidney function tests) and also a contrast-enhanced MRI within 2 days prior to sonication. Patients discontinued the study if tumor progression was identified according to the Response Assessment in Neuro-Oncology (RANO) criteria[51]. For patients continuing the study, a subsequent MRI exam was to be performed immediately after LIPU/MB treatment.

A 3.0 T MRI was used for all imaging exams at each site. At each exam, standard FLAIR, T1-weighted contrast-enhanced, SWAN, SWI, and diffusion sequences were obtained. T1-weighted MR images were analyzed to grade the type and extent of BBBD for post-sonication images and for tumor evolution in pre-sonication images. The gadolinium agent used was dependent on the site, and the following macrocyclic agents were allowed on protocol and were administered according to their respective labels: Dotarem® (gadoterate meglumine), Gadavist® (gadubutrol), or Prohance® (gadoteridol). To limit the exposure to gadolinium agents, post-MRI procedures were performed only at Cycles 1, 2, and 3. Pre and post-ultrasound T2*/SWI images were centrally reviewed by a neuroradiologist (BPL) with an American Board of Radiology certificate of added qualification in diagnostic neuroradiology and with more than 15 years of clinical experience as an attending neuroradiologist.

## BBB opening assessment on MRI

The effectiveness of BBB opening with the SC9 was assessed centrally by comparison of gadolinium-enhanced MR images acquired before and after ultrasound sessions. The analysis was performed using the automated image processing pipeline according to the algorithm previously published[38]. In this grading analysis, emitters in front of the resection cavity and residual hyperintense tumor were excluded. Relative gadolinium enhancement maps from pre- to post-sonication images were computed after bias correction, brain segmentation[52], normalization, and non-rigid registration[53]. Sonicated regions of interest (ROI) were defined by 10-mm diameter × 75-mm length cylinders in front of each of the 9 emitters of the implant, considering only brain tissue that was not enhanced prior to sonication. The volume with detectable ultrasound-induced gadolinium enhancement was determined in the sonicated ROI by thresholding the enhancement map (threshold level: 1st centile of non-sonicated control ROI). A BBBD grade was automatically assigned to each emitter with enhanced volume >0.5 mL using the 0–3 scale defined in ref. 28: grade 0–1 for enhancement in the subarachnoid space; otherwise, grades 2–3 was assigned (enhancement in gray or/and white matter).

Gadolinium enhancement attributable to BBB opening induced by the SC9 implant was also qualitatively evaluated as previously described[33,38]. Enhancement maps were computed from non-rigidly registered pre- and post-sonication T1w images. A sonicated region of interest (ROI) was defined by nine 10-mm diameter × 75-mm length cylinders localized in front of the emitters of the SC9 implant seen on the MRI. For the BBB closure analysis, the 90th percentile of relative enhancement intensity in the sonicated ROI was computed as an intensity metric.

## Contrast-enhancing tumor progression assessment on MRI

To evaluate the local effect of the treatment, an analysis was performed using the monthly pre-ultrasound and end-of-study MR images. The T1w contrast-enhancing tumor-related region segmented with a semi-automatic method (ITK Snap)[54]. The total volume of this hyperintense region was evaluated at each treatment cycle in the whole brain, and in the region targeted by the implant, considering nine 20x80mm cylinders in front of the emitters. This region corresponded to 10 × 75 mm cylinders with an additional diffusion margin of 5 mm[34]. As a metric to evaluate tumor control during the treatment period, the slope of enhancing tumor volume was calculated for each patient from the first sonication to the end of treatment and expressed in mL/month.

To further evaluate tumor progression likelihood as a function of distance to the region targeted by the emitters, the difference of this hyperintense volume between the last MRI and the first cycle was used as a progression mask. The distribution of the volume of this progression mask relative to the distance to the axes of the nine emitters was computed.

## Statistics and reproducibility

For the Phase 1, a standard 3 + 3 escalation design was used. For Phase 2, the number of patients was calculated according to the criteria below. A BBB opening success was defined by a T1W contrast enhancement in gray matter (grade 2) or in gray and white matter (grade 3) on two-thirds of the emitters. The proportion ($\pi$) of ultrasound sessions that were classified as being successful in opening the BBB was compared to an arbitrary objective performance criterion (OPC) of 0.30 at the significance level of a one-sided 2.5%. With a true proportion of success of 0.70 and an OPC of 0.30, a sample size of 15 patients (Cohort C) with 1 ultrasound session was estimated to provide a statistical power of 86%. The BBB opening effectiveness was demonstrated if the lower limit of the 95% CI was higher than 0.30. No statistical tests were planned for Cohort D. Descriptive statistics were used as applicable to summarize the study data unless otherwise specified. Statistical analysis of trial data was performed by an external statistician (EG), except for image analysis of MR data, which was performed by GB. No data were excluded from the analysis. The trial participants were not randomized. The Investigators were not blinded to allocation during experiments and outcome assessment.

**Reporting summary**

Further information on research design is available in the Nature Portfolio Reporting Summary linked to this article.

## Data availability

Aggregated data and associated supporting documents (e.g., protocol) will be made available upon request. Individual particant data that underlie the results reported in this article, after deidentification, will be shared upon request after publication and ending 36 months following article publication to researchers who provide a methologically sound proposal. Proposals should be directed to the corresponding author, Alexandre Carpentier. All remaining data can be found in the Article, Supplementary, and Source Data files. Source data are provided with this paper.

## Code availability

The code used for analysis of data during the current study are available from the corresponding author upon request.

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

## Acknowledgements

We thank the patients and their families for agreeing to participate in this trial. The authors also acknowledge the members of the independent data monitoring committee: Martin van den Bent, Veronica Chiang, and Charles Cobbs. Special thanks to the nurses and the clinical research teams for their collaboration at all the participating sites, in particular Mélanie Bourgoin, Nabila Rousseaux, Armelle Rametti, and Haysam Salman (all at APHP, Paris), and Christina Amidei, Karyn Schmidt, Rachel Ward and Gianna Mirabelli (at Northwestern University, Chicago), as well as Didier Autran, Sébastien Boisseauneau (at La Timone Hospital, Marseille), and Cécile Trouba and Cécile Novello (at P. Wertheimer Hospital, Lyon). We also thank Lantheus for supplying DEFINITY® for this trial. Collection and analysis of study data were performed by an independent contract research organization as well as by an independent statistician (EG). Carthera planned the study design, performed analysis of imaging data, and aided in manuscript preparation, as detailed in the Author Contributions Statement.

## Author contributions

A.C., C.D., and A.I. conceived and designed the trial. A.C., R.S., A.M.S., H.D., O.C., B.M., F.D., J.G., N.B., P.M., J.G., J.W., and A.I. were investigators for the trial and participated in patient care related to trial procedures and assessments. E.G. contributed to the statistical design and analysis of the trial data. B.L., G.B., C.S., and M.C. contributed to MRI analysis. A.C., A.I., C.D., and C.S. accessed and confirmed the accuracy of the results presented in the manuscript. A.C., R.S., A.M.S., C.D., C.S., G.B., M.C., and AI participated in the writing and revision of the manuscript.

## Competing interests

AMS and RS have received in-kind (drug) support from Bristol-Myers Squibb, in-kind (ultrasound devices) and research support from Carthera, and in-kind (drug) and research support from Agenus. AMS and RS are co-authors of intellectual property filed by Northwestern University related to therapeutic ultrasound. RS has acted or is acting as a scientific advisor or has served on advisory boards for the following companies: Alpheus Medical, AstraZeneca, Boston Scientific, Carthera, Celularity, GT Medical, Insightec, Lockwood (BlackDiamond), Northwest Biotherapeutics, Novocure, Syneos Health (Boston Biomedical), TriAct Therapeutics, and Varian Medical Systems. AMS is a consultant for Carthera and Enclear Therapeutics. MC, CD, CS, GB, and AC are employees of Carthera, inventors of patents related to the technology, or have stock ownership in Carthera. AC has received funding support from Horizon 2020 European Innovation Council; is a paid consultant of Carthera; and is part of the Board of Directors of Carthera. FD is acting as a scientific advisor or has served on advisory boards for the following companies: Novocure, Servier. AI has received research grants from Carthera, Transgene, Sanofi, Nutritheragene; travel funding from Enterome and Carthera; personal fees for advisory board from Leo Pharma, Novocure, Novartis, and Boehringer Ingelheim outside the submitted work. The remaining authors declare no competing interests.

## Additional information

[1]Sorbonne Université, AP-HP, Hôpitaux Universitaires La Pitié Salpêtrière - Charles Foix, Service de Neurochirurgie, Paris, France. [2]Department of Neurological Surgery, Feinberg School of Medicine, Northwestern University, Chicago, IL, USA. [3]Northwestern Medicine Malnati Brain Tumor Institute of the Lurie Comprehensive Cancer Center, Feinberg School of Medicine, Northwestern University, Chicago, IL, USA. [4]Aix-Marseille Univ, APHM, CNRS, INP, Inst Neurophysiopathol, CHU Timone, Service de Neuro-Oncologie, Marseille, France. [5]Hospices Civils de Lyon, Université Claude Bernard Lyon 1, Service de Neuro-Oncologie, Hospices Civils de Lyon, Cancer Research Center of Lyon, INSERM U1052, CNRS UMR 5286, Cancer Cell Plasticity Department, Lyon, France. [6]CHU Angers, Angers, France. [7]Departments of Neurology and Neurosurgery, University of California, San Francisco, CA, USA. [8]Department of Neurosurgery, The University of Texas MD Anderson Cancer Center, Houston, TX, USA. [9]Departments of Radiology and Radiation Oncology, Feinberg School of Medicine, Northwestern University, Chicago, IL, USA. [10]Biossec, Paris, France. [11]Carthera, Lyon, France. [12]Sorbonne Université, Inserm, CNRS, UMR S 1127, Institut du Cerveau, ICM, AP-HP, Hôpitaux Universitaires La Pitié Salpêtrière - Charles Foix, Service de Neuro-Oncologie, Paris, France. ✉e-mail: alexandre.carpentier@aphp.fr

