## [Peer Review File · Nature Communications]

Reviewers' Comments:

Reviewer #1:

Remarks to the Author:

The authors developed a nine-emitter skull implantable ultrasound device for repetitive blood brain barrier opening using focused ultrasound and microbubbles. They demonstrate repetitive LIPU/MB to transiently disrupt the BBB in post-operative recurrent glioblastoma patients. The SC9 device is accessed transcutaneously using a needle which transmits energy to create sequential pulses from each emitter (1 MHz, 25 ms pulse, 0.5 Hz, 270 seconds). They suggest that FUS-BBB opening plus carboplatin administration was well-tolerated and may increase the effectiveness of systemic drug therapies, but provide little/no evidence for enhanced drug delivery. There were device and treatment complications including wound infection and mostly (97%) experienced common terminology criteria for adverse events (CTCAE) (Table 2)

Some comments and requested modifications:

1. The mean time of treatment was 9.9 +/- 3.4 min. How was this determined?
2. What were the ultrasound power settings for each treatment and how were these determined?
3. There is no longitudinal imaging or T2*/SWI imaging given
4. The authors use the term 'large' opening. What does this mean and /or what is this in reference to?
5. How was the % new enhancement determined, especially given the imaging and contrast differences across the trial sites (Figure 4)? Is the new enhancement normalized against other contrast enhancing structures, such as the choroid plexus?
6. T1C imaging is non-specific and cannot be used to gauge treatment responses or recurrence. There is no consideration of 'treatment effect' vs. 'tumor recurrence'

Reviewer #2:

Remarks to the Author:

In the study titled, "Temporary Blood-Brain Barrier Disruption Using a Nine-Emitter Skull Implantable 2 Ultrasound Device in Patients with Recurrent Glioblastoma Undergoing Carboplatin 3 Chemotherapy" by Carpentier and colleagues, they performed a single arm Phase I/II multi-institutional study to test the safety and efficacy of blood brain barrier (BBB) opening for patients with recurrent glioblastoma (GBM) while delivering carboplatin (a drug that typically has poor CNS penetrance). Using their new 9-transducer device, they performed a MTD study to determine the highest active beam level. This is a novel study that addresses an historical problem when treating brain tumors that have diffuse infiltration and the authors and institutes should be commended for performing this complex collaborative study to advance the field of ultrasound and GBM.

This study has major significance in the field. To date, there are at least 3 other ultrasound devices that are currently being investigated. The sonocloud device utilizes an implantable device which is unique and allows for easy access for ultrasound delivery in combination with drug uptake. This platform can be adapted across multiple institutes and has the potential to change current treatment paradigm. Lastly, the degree of BBB opening achieved is considered one of the largest to date and was reportedly relatively safe. This helps advance the understanding and potential of using sound waves to open the BBB.

The experimental design and study design is sound. The conclusions drawn from this study are appropriate and the work can be reproduced.

Major Comments:

- 1.) One death was noted in the study. This occurred prior to sonication, but occurred post-op. Per authors, it was likely from PE. It's unclear how PE was determined to be cause of fatality and it is not with 100% certainty that it was unrelated to the surgical procedure. Also unclear whether anticoagulation was used and whether implantation of device limited use of PE prophylaxis.
- 2.) Was there further analysis to see how the quality of the BBB opening affected treatment

response in terms of PFS/OS.

Minor Comments:

- 1.) In the discussion, the authors referred to Sonabend et al. 2022 Submitted. I believe this paper is published, update the reference
- 2.) Did the author perform MRI scans for any of the patients to determine the BBB closure time?
- 3.) Volumetric analysis of the BBB opening achieved in 3 transducer, 6 transducer and 9 transducer.
- 4.) Did the size/efficiency of BBB opening correlate with treatment response and outcomes
- 5.) What was the mOS for patients with 3 and 6 emitters and what degree of BBB opening in relationship to tumor was covered
- 6.) Any analysis of location of implant placement and efficiency of BBB opening
- 7.) Is there any quantifying data looking at tumor size status post treatment.

Reviewer #3:

Remarks to the Author:

This study reported a median OS of 12 months and median PFS of 2.5-3 months using the combination of SonoCloud-9 system with monthly given carboplatin in recurrent GBM patients. The procedure was reported to be safe and a dose-escalation to maximal 9 emitters design was reported. An additional 24 patients underwent 9 emitters-on repeated therapy demonstrated grade 2-3 BBBD were observed in 90% of the activated emitters. Furthermore, a different cohort D was designed to test that carboplatin was given immediately after BBBD, which showed the radiological outcome seemed better than in the cohort that BBBD was performed within 60 minutes after carboplatin infusion, suggesting the timing of drug administration with BBBD should be optimized in future studies.

Generally, this study is well designed and the report is concise and comprehensive. I have few suggestions or questions:

1. In the methodology section, the patients were recruited if disease recurred after TMZ-CCRT. Could the authors provide a supplementary summary of which treatments these patients received before they entered into this trial? How many percentage of the patients failed after first line CCRT and adjuvant TMZ, how many after 2nd line bevacizumab, how many after 2nd line chemotherapy?
2. 6 patients (18%) complained of severe pain after connection to sonication, did the procedures continued or stopped? What is the possible mechanism of this pain response, was is related to needle puncture or anything related to the emission of the device? Could the authors explain more in the discussion?
3. For the clinical outcomes in results, since the mOS in cohort D was similar to that in cohort C, it would be inappropriate to claim that "a closer time of administration of carboplatin to sonications led to improved radiological and "clinical outcomes".

• What are the noteworthy results?

1. the actual BBB closure time was estimated to be 1.3 hours after BBBD.

• Will the work be of significance to the field and related fields? How does it compare to the established literature? If the work is not original, please provide relevant references.

Yes, it provided further evidence on using BBBD combined with carboplatin in rGBM patients.

• Does the work support the conclusions and claims, or is additional evidence needed?

Yes.

• Are there any flaws in the data analysis, interpretation and conclusions? Do these prohibit publication or require revision?

Yes, the data provided support the conclusions.

• Is the methodology sound? Does the work meet the expected standards in your field?

Yes, the methodology is sound.

- Is there enough detail provided in the methods for the work to be reproduced?

Yes.

Reviewer #4:

Remarks to the Author:

The authors have found some interesting and encouraging findings on their phase I/II trial. However, the results are quite preliminary and the authors are making some claims that are not supported by the evidence provided in the trial. Many of the important comments below are suggested areas where there are places where the interpretations/conclusions are not supported by the data. These need to be revised. Note that not all instances where this occurs are indicated in the comments. The comments also raise other suggestion for improvement in the manuscript.

Suggestions:

- (1) Please provide 95% confidence intervals (or some other percent if specified a priori in the protocol) for all point estimates, including those provided in the abstract.
- (2) Do not use the term "trend" as in "... with a trend of improved tumor control and survival ...", which is found in the abstract. The trend is not statistically significant and one group will have a higher value than another group (ties are very rare) just due to chance. No evidence that supports the use of the word "trend".
- (3) Throughout the manuscript the term "rate" (or percent) is missing when providing values at an indicated time point. For example it should be the 1-year OS RATE was 52% rather than the 1-year OS was 52%.
- (4) The last sentence of the abstract conclusion needs to be revised. It implies that this study improved tumor control and efficacy, which this data did not show.
- (5) Why was carboplatin used rather than lomustine, which is generally the drug used in recurrent glioma. The rationale for using carboplatin was a bit weak. Is it because this technology only works with IV administered drugs? If this is the case, should state that needed to use an IV administered drug. What does the fact that drugs would need to be IV (if this is the case) mean moving forward? (might be addressed in discussion)
- (6) It seems that the purpose of this study was to evaluate safety and efficacy of transient disruption of the BBB IN CONJUNCTION of treatment with carboplatin. The use with carboplatin was not explicitly stated. (It might be the case there would need to be small safety studies of some sort with the use of the technology with other drugs similar for the need of small safety studies of RT combined with other known drugs not previously used with RT.)
- (7) For the PFS and OS endpoint, what patients were censored and at what timepoints?
- (8) Why is the pre-surgery tumor size limited to 70 mm in diameter or less? This is not a usual eligibility criterion for recurrent GMB studies.
- (9) The evaluation of tumor progression is not clear. Why wasn't RANO criteria used. Tumor volume is not an established method for determining tumor progression. What increase in tumor volume was deemed a tumor progression? What about patients who had no evidence of disease after their resection?
- (10) What is an objective performance criterion that is used for the proportion of ultrasound sessions that were classified as being successful in opening the BBB? What is the basis of using 0.30 as an historical control value? There is a level of significance provided but no sample size, power, or detectable difference.
- (11) What was the a priori determined sample size for cohorts C and D? What was the basis of these? There is no real sample size justification provided. What is provided is incomplete (just a rule is given for what determines effectiveness but no indication of what sample size is needed to determine effectiveness under an alternative hypothesis).
- (12) In general, this really does not seem to have a phase II component. Phase II is generally used to determine a signal of potential efficacy. BBB disruption is not known to be a surrogate for clinical efficacy. In theory, I am okay with calling this a phase IIa but might be better stated as a feasibility study of the technology given a traditional phase II endpoint is not being used as the primary endpoint.
- (13) In the Table 1 title/legend, please remove the statement that says no significant differences between cohorts C and D. The study is not powered to determine differences. Readers can use

their judgement as to whether there are potentially meaningful differences between the cohorts. Stating no significant difference implies there is power to detect meaningful differences, which there is not.

(14) What is the average time for surgical resection without the implantation in a similar patient population (tumors less than 70 mm) for a comparison as to how much extra time the surgery with implantation required. Just stating the average surgical time for the patients does not really provided information regarding the additional time the implantation took (versus if there was a surgery without intent to implant).

(15) There is concern about using BBBD as an indication of efficacy, especially since it is not a validated surrogate of clinical benefit and because of the demonstrated variability in enhancement among the institutions. It is not clear how reliable/reproducible this endpoint is.

(16) How well were the institutions balanced in cohorts C and D. The concern is that if not balanced, differences in outcomes between the cohorts could be confounded by differences in the institutions.

(17) Please clarify exactly how tumor control rate was computed.

(18) There is concern that tumor control rate is impacted by the amount of residual tumor after resection. Were the groups balanced on this? It appears as though there might have been differences in baseline mean/median values of hyperintense T1w in implant ROI between the cohorts.

(19) Rather than using --- when a value cannot be estimated, please indicate "not estimable".

(20) Please include 95% confidence intervals in Table 3.

(21) Cannot say that the media OS compares favorably to historical recurrent GBM trial/studies because this is a very select patient population compared to most other recurrent GBM studies. Patients with resectable recurrent tumors as well as recurrent tumors less than 70 mm will have better clinical outcomes than the general recurrent GMB population.

(22) There is no evidence that cohort D has better clinical outcomes compared to cohort C. By chance, one cohorts will have better values than another. However, the p-values do not indicated significant differences AND the confidence intervals completely overlap for all the clinical outcomes. At best, the findings might support an hypothesis that needs to be tested but there is no evidence that one cohort is better than the other in terms of clinical outcomes.

The manuscript needs to be revised to remove misleading comments/implications that cohort D has better clinical outcomes than cohort C. Only place it might be better is in the BBBD effectiveness (which is not a clinical outcome) and perhaps tumor control, which is not measured with a validated measurement.

RESPONSE TO REVIEWERS FOR NCOMMS-23-08292-T

Reviewer #1 - Ultrasound, BBB drug delivery, trials (Remarks to the Author):

The authors developed a nine-emitter skull implantable ultrasound device for repetitive blood brain barrier opening using focused ultrasound and microbubbles. They demonstrate repetitive LIPU/MB to transiently disrupt the BBB in post-operative recurrent glioblastoma patients. The SC9 device is accessed transcutaneously using a needle which transmits energy to create sequential pulses from each emitter (1 MHz, 25 ms pulse, 0.5 Hz, 270 seconds). They suggest that FUS-BBB opening plus carboplatin administration was well-tolerated and may increase the effectiveness of systemic drug therapies but provide little/no evidence for enhanced drug delivery. There were device and treatment complications including wound infection and mostly (97%) experienced common terminology criteria for adverse events (CTCAE) (Table 2).

Some comments and requested modifications:

1. The mean time of treatment was 9.9 +/- 3.4 min. How was this determined?

This time (from needle connection to the needle disconnection) was collected and reported by clinical sites participating in the study on the eCRF for the trial. Note that the total sonication time is 270 seconds (4.5 minutes) and this duration is already specified in the Methods. An additional sentence was added to the Methods to clarify the mean treatment time for the reader:

"Timing and duration of implantation and sonication procedures was collected on the eCRF from sites participating in the trial."

In addition, this point was further clarified in the Results:

"The mean overall duration of the sonication procedure was 9.9 minutes, which includes the time to make the needle connection to the implant and the 4.5 min sonication procedure [SD: ± 3.38 min]."

2. What were the ultrasound power settings for each treatment and how were these determined?

The ultrasound power settings were determined by the generator for each implant. The SonoCloud-9 implant contains a memory chip embedded in the device that has a calibration coefficient that is unique for each ultrasound emitter and that was determined during the device fabrication process. The ultrasound power was designed so that the acoustic output in water had a peak pressure amplitude of 1.03 MPa. An additional clarification was added to the Methods to clarify this point:

"The peak pressure amplitude generated by each ultrasound emitter of the SonoCloud-9 device, which were pre-calibrated before implantation, was set to 1.03 MPa in water. A unique calibration coefficient for each emitter was read by the SonoCloud generator to set the electrical power sent at activation of each emitter."

3. There is no longitudinal imaging or T2*/SWI imaging given

We agree that no information was presented on these images in the article. We performed additional analysis to respond to these comments and pre-sonication and post-sonication registered SWI/T2* image pairs acquired for 54 treatments (33 patients) were centrally reviewed by a neuroradiologist (B.P.L.) with an American Board of Radiology (ABR) certificate of added qualification (CAQ) in diagnostic neuroradiology with more than 15 years of clinical experience as an attending neuroradiologist. We've added additional text in the Methods, Results, and Discussion about this point and added B.P.L as a co-author to the manuscript. The modifications to the text are below:

Methods:

"Pre and post ultrasound T2/SWI images were reviewed by a neuroradiologist (B.P.L.) with an American Board of Radiology (ABR) certificate of added qualification (CAQ) in diagnostic neuroradiology and with more than 15 years of clinical experience as an attending neuroradiologist."*

Results:

"A total of 52 pre and post ultrasound T2/SWI pairs were reviewed by a trained neuro-radiologist (BPL), corresponding to 52 sonications in 33 patients. In 36/52 of the images (69%), at least one or more pre-existing hypointense regions were present in the sonication field. In all cases, these pre-existing hypointense SWI signal abnormalities remained similar in the post-US image. A single new hypointense SWI signal abnormality (<5 mm in diameter) was observed in the sonication field in post-US SWI images for 6/52 treatments (11%). In all cases where a later follow-up SWI image was available (i.e. following months), no hypointense signal increase was observed."*

Discussion:

"New small (<5 mm) hypointense regions were observed in 6/52 treatments when comparing available pre and post sonication T2/SWI image pairs. In routine clinical practice, none of these would have been considered to be clinically significant due to their small size and lack of mass effect according to a trained neuro-radiologist (BPL) in the setting of postsurgical MRI follow up after a neurosurgical tumor resection or after any type of chemotherapy or radiation therapy. In a recent study using a transcranial ultrasound system to disrupt the BBB, new T2* changes were observed in 50% of the targeted regions after BBB disruption, but no evidence of tissue damage was observed following surgical removal of this tissue 2 to 4 hours following treatment. The authors furthermore proposed such changes may be due to local gadolinium or protein extravasation (52). These SWI features are also frequently observed in GBM patients due to radiosurgery or tumor angiogenesis and have been proposed as a way to distinguish high grade gliomas from low grade ones (53)."*

4. The authors use the term 'large' opening. What does this mean and /or what is this in reference to?

We used this term in comparison to our previous clinical trials with the SonoCloud-1 device where the BBB opening targeted area was 1/9th the volume of that targeted in this work. This term has been further clarified in the text when this intended meaning was not clear.

5. How was the % new enhancement determined, especially given the imaging and contrast differences across the trial sites (Figure 4)? Is the new enhancement normalized against other contrast enhancing structures, such as the choroid plexus?

The % enhancement shown in Figure 4B was determined by comparing images obtained after BBB opening using the SonoCloud with images obtained 1-2 days before the sonication procedure. The images were not normalized against a contrast enhancing structure, but instead were normalized against a pre-sonication image. Additional text has been added to the Methods to provide further information to the reader:

"Gadolinium enhancement attributable to BBB opening induced by the SC9 implant was evaluated as previously described (Asquier et al. 2019). Enhancement maps were computed from non-rigidly registered pre- and post-sonication T1w images. A sonicated region of interest (ROI) was defined by nine 10-mm diameter x 75-mm length cylinders localized in front of the emitters of the SC9 implant seen on the MRI. For the BBB closure analysis, the 90th percentile of relative enhancement intensity in the sonicated ROI was computed as an intensity metric."

6. T1C imaging is non-specific and cannot be used to gauge treatment responses or recurrence. There is no consideration of 'treatment effect' vs. 'tumor recurrence'

We agree with the reviewer that T1 contrast-enhancement can be non-specific, though most trials do look at growth of nodular contrast enhancement as a surrogate for disease progression. Nevertheless, we presented this data for every patient in Cohorts C and D in Figure 6A. In patients who showed evidence of treatment response, we tended to observe a slight increase in T1C up to the 2nd cycle, followed by a decrease. In patients who showed no effect of treatment, the change in T1C was evident and tended to increase quickly until the

patient progressed off the trial. The correlation between these T1C changes and standard clinical efficacy endpoints will be further examined in a future Ph3 clinical trial that is being planned.

Reviewer #2 - Glioblastoma clinical trials (Remarks to the Author):

In the study titled, "Temporary Blood-Brain Barrier Disruption Using a Nine-Emitter Skull Implantable 2 Ultrasound Device in Patients with Recurrent Glioblastoma Undergoing Carboplatin 3 Chemotherapy" by Carpentier and colleagues, they performed a single arm Phase I/II multi-institutional study to test the safety and efficacy of blood brain barrier (BBB) opening for patients with recurrent glioblastoma (GBM) while delivering carboplatin (a drug that typically has poor CNS penetrance). Using their new 9-transducer device, they performed a MTD study to determine the highest active beam level. This is a novel study that addresses an historical problem when treating brain tumors that have diffuse infiltration and the authors and institutes should be commended for performing this complex collaborative study to advance the field of ultrasound and GBM.

This study has major significance in the field. To date, there are at least 3 other ultrasound devices that are currently being investigated. The SonoCloud device utilizes an implantable device which is unique and allows for easy access for ultrasound delivery in combination with drug uptake. This platform can be adapted across multiple institutes and has the potential to change current treatment paradigm. Lastly, the degree of BBB opening achieved is considered one of the largest to date and was reportedly relatively safe. This helps advance the understanding and potential of using sound waves to open the BBB.

The experimental design and study design is sound. The conclusions drawn from this study are appropriate and the work can be reproduced.

Major Comments:

1.) One death was noted in the study. This occurred prior to sonication, but occurred post-op. Per authors, it was likely from PE. It's unclear how PE was determined to be cause of fatality and it is not with 100% certainty that it was unrelated to the surgical procedure. Also unclear whether anticoagulation was used and whether implantation of device limited use of PE prophylaxis.

The Pulmonary embolism (PE) was reported as the cause of death. Symptomatic venous thromboembolic events including deep vein thrombosis and PE are frequent in patients with glioblastoma as a surgical complication and the event was considered as a post operative complication in this patient. There was no reason to believe that the implantation portion of the procedure contributed to the development of this PE. Low molecular weight heparin was prescribed. This point was further clarified in the text:

"One fatal event (pulmonary embolism) occurred 7 days post-surgery, but this patient did not receive any sonications/activation of the device due to the event happening prior to initiating cycle 1 and was not considered as related to the SonoCloud-9 implantation procedure as these events are frequently reported complications in GBM patients (45,46)."

2.) Was there further analysis to see how the quality of the BBB opening affected treatment response in terms of PFS/OS.

No further analysis investigating the correlation BBB opening grading and clinical outcome was performed.

Minor Comments:

1.) In the discussion, the authors referred to Sonabend et al. 2022 Submitted. I believe this paper is published, update the reference

We've updated the reference in the text as indeed this article was published in 2023.

2.) Did the author perform MRI scans for any of the patients to determine the BBB closure time?

We performed an analysis of BBB closure time in this work and also recently reported this in the Lancet Oncology article by Sonabend et al. 2023. The analysis of BBB closure time was an ad hoc analysis that was not pre-planned and was based on the availability of timing for MRI scans after BBB opening in the patients treated but

was repeated on different patient cohorts in this study and the study by Sonabend et al. 2023. This analysis is shown in Figure 4B and indicates a BBB closure half-life of 1.3 hours. These results are complementary/supportive and show the reproducibility of the BBB closure dynamics. An additional sentence was added to the Discussion:

"These results are in agreement with those recently published in a separate patient cohort treated with the SonoCloud-9 in which there was rapid restoration of the BBB within 1 hr of sonication (33)."

3.) Volumetric analysis of the BBB opening achieved with 3 transducer, 6 transducer and 9 transducer.

This analysis was conducted but is not reported in the manuscript due to the low number of patients with 3 emitters activated (3 patients, 2 sites, 6 post-treatment T1w images available) or 6 emitters activated (3 patients, 1 site, 7 post-treatment T1w images).

The overall volume with post-sonication T1w enhancement increased with number of activated elements (ANOVA: $p=0.005$). On the other hand, there is no significant difference in enhancement intensity in the targeted brain when comparing treatments with 3, 6 or 9 emitters activated ($p=0.588$). This suggests that there is no saturation effect or plateau in BBB-disruption efficiency when increasing targeted volume, but this data is insufficient to strongly support this conclusion.

4.) Did the size/efficiency of BBB opening correlate with treatment response and outcomes

We looked for potential correlations between overall survival or tumor growth rate and various metrics of Gd enhancement related to ultrasound-mediated BBB disruption (enhancement intensity and volume). No correlation was found between treatment response or outcome and the BBB opening metrics evaluated on post-Gd post-Sonication T1w images. Note that this analysis is hindered by the fact that BBB enhancement metrics are biased by variability of delays between sonication and Gd injection, and by the type of contrast agent used at sites. We plan to continue to look at these potential correlations though in future trials.

5.) What was the mOS for patients with 3 and 6 emitters and what degree of BBB opening in relationship to tumor was covered

Median OS for patients with 3 and 6 emitters was not estimated due to low number ($n=3$) of patients. The degree of BBB opening in relation to the tumor covered will be addressed in future work.

6.) Any analysis of location of implant placement and efficiency of BBB opening

These are great points and things that we plan to look at in the larger clinical trial that we are planning when there are more evaluable patients.

7.) Is there any quantifying data looking at tumor size status post treatment.

No MRIs were collected after treatment discontinuation.

Reviewer #3 - Ultrasound, BBB drug delivery, trial (Remarks to the Author):

This study reported a median OS of 12 months and median PFS of 2.5-3 months using the combination of SonoCloud-9 system with monthly given carboplatin in recurrent GBM patients. The procedure was reported to be safe and a dose-escalation to maximal 9 emitters design was reported. An additional 24 patients underwent 9 emitters-on repeated therapy demonstrated grade 2-3 BBBD were observed in 90% of the activated emitters. Furthermore, a different cohort D was designed to test that carboplatin was given immediately after BBBD, which showed the radiological outcome seemed better than in the cohort that BBBD was performed within 60 minutes after carboplatin infusion, suggesting the timing of drug administration with BBBD should be optimized in future studies.

Generally, this study is well-designed, and the report is concise and comprehensive. I have few suggestions or questions:

1. In the methodology section, the patients were recruited if disease recurred after TMZ-CCRT. Could the authors provide a supplementary summary of which treatments these patients received before they entered into this trial? How many of the patients failed after first line CCRT and adjuvant TMZ, how many after 2nd line bevacizumab, how many after 2nd line chemotherapy?

The majority of patients (88%) were enrolled at first recurrence of the disease except one (in cohort A) at third recurrence (3%) and three (in cohorts B and C) at second recurrence (9%). This information is given for Cohorts C and D in Table 1.

Across all cohorts, all participants received chemotherapy prior to enrollment, with 24% of participants (8 participants) reported with a complete response (CR), 6% (2 participants) with a partial response (PR), 41% (14 participants) with stable disease, and 29% (10 participants) with progressive disease (PD) as the best overall response to chemotherapy. All participants received prior radiotherapy. A total of 59% of participants had prior total tumor resection and 32% had partial tumor resection.

All patients received temozolomide as first line chemotherapy treatment. In addition, one patient received TMZ in combination/sequential regimen with pembrolizumab, 1 with marizomib and 1 patient with nivolumab. For the patients enrolled at second and third recurrence, previous treatments included lomustine and lomustine and temozolomide respectively. None of the patients received bevacizumab.

2. 6 patients (18%) complained of severe pain after connection to sonication, did the procedures continued or stopped? What is the possible mechanism of this pain response, was it related to needle puncture or anything related to the emission of the device? Could the authors explain more in the discussion?

The procedure did continue after these reports of severe pain. The patients complained of some pain at time of the transdermal needle puncture. There are different reasons for this pain, such as sensitivity of the patients to pain, an anesthetic cream not applied sufficiently in advance of the connection, sensitivity of the wound area after surgery, or thickness/resistance of the skin and or galea requiring more pressure to insert the needle. No pain was reported due to the sonication itself. These reported severe pain events were very transient with a mean duration of 5 minutes (1-10 minutes). These severe pain events were only reported in 6/34 patients and 7/90 (7.8%) needle connection procedures. Additional text has been added to the revised manuscript to clarify this point:

"Although 6/34 patients reported severe pain, these events were due to the needle puncture at time of sonication with a mean duration of 5 minutes and did not prevent the patient from receiving treatments and were not due to the ultrasound emission itself. Additional steps, including use of analgesic creams prior to the needle connection procedure have been incorporated into on-going and future clinical trials with the SonoCloud device to minimize patient discomfort."

3. For the clinical outcomes in results, since the mOS in cohort D was similar to that in cohort C, it would be inappropriate to claim that "a closer time of administration of carboplatin to sonications led to improved radiological and "clinical outcomes".

We have further revised the Discussion and Conclusions of the manuscript in response to the reviewer's comments.

Reviewer #4 - Biostatistics, clinical trial design (Remarks to the Author):

The authors have found some interesting and encouraging findings on their phase I/II trial. However, the results are quite preliminary, and the authors are making some claims that are not supported by the evidence provided in the trial. Many of the important comments below are suggested areas where there are places where the interpretations/conclusions are not supported by the data. These need to be revised. Note that not all instances where this occurs are indicated in the comments. The comments also raise other suggestion for improvement in the manuscript.

Suggestions:

(1) Please provide 95% confidence intervals (or some other percent if specified a priori in the protocol) for all point estimates, including those provided in the abstract.

These have been added in the revised manuscript.

(2) Do not use the term "trend" as in "... with a trend of improved tumor control and survival ...", which is found in the abstract. The trend is not statistically significant, and one group will have a higher value than another group (ties are very rare) just due to chance. No evidence that supports the use of the word "trend".

Thank you. We've removed the word "trend" and modified the abstract to comply with the 150 word limit and instead only included the results from the patient cohort that received carboplatin before sonication in the abstract.

(3) Throughout the manuscript the term "rate" (or percent) is missing when providing values at an indicated time point. For example it should be the 1-year OS RATE was 52% rather than the 1-year OS was 52%.

Thank you, we've added the word "rate" throughout the revised manuscript where it was missing.

(4) The last sentence of the abstract conclusion needs to be revised. It implies that this study improved tumor control and efficacy, which this data did not show.

We've furthermore revised the abstract to fit the 150 word limit and modified the wording around tumor control and efficacy.

(5) Why was carboplatin used rather than lomustine, which is generally the drug used in recurrent glioma. The rationale for using carboplatin was a bit weak. Is it because this technology only works with IV administered drugs? If this is the case, should state that needed to use an IV administered drug. What does the fact that drugs would need to be IV (if this is the case) mean moving forward? (might be addressed in discussion)

Both IV and oral drugs can be used with BBB opening and are currently being used in clinical protocols, with oral TMZ being used in on-going clinical trials by Carthera and Insightec.

We've added an additional sentence to the Introduction to address the reviewer's comment:

"The only other drugs that are commonly used for treatment of GBM, lomustine and temozolomide, have a reported brain/plasma ratio of 20-40% (31)."

The choice of drug in our work was to intentionally use a drug that does not cross the BBB well but that has showed some signed of efficacy and to demonstrate if significantly enhancing the brain concentration of the drug could enhance efficacy.

(6) It seems that the purpose of this study was to evaluate safety and efficacy of transient disruption of the BBB IN CONJUNCTION of treatment with carboplatin. The use with carboplatin was not explicitly stated. (It might be the case there would need to be small safety studies of some sort with the use of the technology with other

drugs similar for the need of small safety studies of RT combined with other known drugs not previously used with RT.)

Yes could be in conjunction to planned carboplatin therapy or as add-on to planned carboplatin therapy. We think that the intention as written is clear for the reader. Indeed, we have on-going studies with other drug therapies such as TMZ and nab-paclitaxel with our SonoCloud device and we are actively investigating the safety of the use of this technology in conjunction with other drug therapies.

(7) For the PFS and OS endpoint, what patients were censored and at what timepoints?

For OS, 3 patients were censored:

- Cohort C-C' 1 patient at 34 months
- Cohort D 2 patients at 18 and 19 months

For PFS, 2 patients were censored:

- Cohort C-C' 1 patient at 6 months
- Cohort D 1 patient at 7 months

(8) Why is the pre-surgery tumor size limited to 70 mm in diameter or less? This is not a usual eligibility criterion for recurrent GMB studies.

The inclusion size to 70 mm in diameter of tumor contrast enhancement was to ensure that the SonoCloud-9 sonication area could cover most of the tumor and peritumoral region.

(9) The evaluation of tumor progression is not clear. Why wasn't RANO criteria used. Tumor volume is not an established method for determining tumor progression. What increase in tumor volume was deemed a tumor progression? What about patients who had no evidence of disease after their resection?

RANO was in fact used to evaluate disease progression by the physician at the clinical sites participating in the protocol. Response rate was not assessed since all patients underwent additional resection and there was often little residual tumor after surgery. Tumor volume was not used as a method of determining tumor progression and was only evaluated after patient's left the trial to evaluate whether there was evidence of slower tumor growth between the cohorts treated.

(10) What is an objective performance criterion that is used for the proportion of ultrasound sessions that were classified as being successful in opening the BBB? What is the basis of using 0.30 as an historical control value? There is a level of significance provided but no sample size, power, or detectable difference.

An objective performance criterion (OPC) refers to a numerical target value defined as an historical control in this single-arm study design. The proportion of successful ultrasound sessions in opening the BBB will be compared to the OPC of 0.30. A performance less than 30% was considered as non-clinically relevant for the device which is intended to allow the crossing of the BBB.

(11) What was the a priori determined sample size for cohorts C and D? What was the basis of these? There is no real sample size justification provided. What is provided is incomplete (just a rule is given for what determines effectiveness but no indication of what sample size is needed to determine effectiveness under an alternative hypothesis).

The number of patients in the expansion phase was determined arbitrarily. The number of patients in the cohort C (12 in addition to the 3 treated in the escalation step) was considered as sufficient to confirm safety observed in the escalation phase, and to test the efficacy according to the BBB opening criteria. In the treatment optimization cohort (cohort D), the same number of patients was enrolled for consistency purpose with cohort C.

(12) In general, this really does not seem to have a phase II component. Phase II is generally used to determine a signal of potential efficacy. BBB disruption is not known to be a surrogate for clinical efficacy. In theory, I am okay with calling this a phase IIa but might be better stated as a feasibility study of the technology given a traditional phase II endpoint is not being used as the primary endpoint.

The reviewer is correct. The Primary endpoints of the present trial were safety and efficacy in terms of BBB opening effectiveness, and feasibility was evaluated as secondary endpoints. It is a phase 1/2a. However, the title of the protocol cannot be changed now.

In "Trial Design", we've modified the text as follows:

"This trial was designed to evaluate the safety of concomitant carboplatin administration with transient disruption of the blood-brain barrier by low intensity pulsed ultrasound (LIPU) using the SonoCloud-9 implantable device as well as the performance of the SonoCloud-9 device to repeatedly disrupt the BBB."

(13) In the Table 1 title/legend, please remove the statement that says no significant differences between cohorts C and D. The study is not powered to determine differences. Readers can use their judgement as to whether there are potentially meaningful differences between the cohorts. Stating no significant difference implies there is power to detect meaningful differences, which there is not.

Ok, we've removed this phrase from the title/legend for Table 1.

(14) What is the average time for surgical resection without the implantation in a similar patient population (tumors less than 70 mm) for a comparison as to how much extra time the surgery with implantation required. Just stating the average surgical time for the patients does not really provided information regarding the additional time the implantation took (versus if there was a surgery without intent to implant).

Added in the revised manuscript:

"The mean estimated additional time required for the device placement is 24 [±13] minutes."

(15) There is concern about using BBBD as an indication of efficacy, especially since it is not a validated surrogate of clinical benefit and because of the demonstrated variability in enhancement among the institutions. It is not clear how reliable/reproducible this endpoint is.

We did not use BBBD as an indication of efficacy in the study, it was only the endpoint of the trial.

(16) How well were the institutions balanced in cohorts C and D. The concern is that if not balanced, differences in outcomes between the cohorts could be confounded by differences in the institutions.

The breakdown of patients at each institution is shown in the table below:

Site	Cohort C (N=15)	Cohort D (N=12)
001	6 (40%)	4 (33%)
002	2 (13%)	0
003	1 (7%)	0
004	4 (27%)	3 (25%)
005	1 (7%)	5 (42%)
006	1 (7%)	0

The majority of the patients were enrolled at Site 001 (Paris) and Site 004 (Chicago). There were more patients enrolled at Site 005 in Cohort D, which could have affected the outcomes, but the breakdown between patients enrolled between France and the USA was similar in both cohorts.

(17) Please clarify exactly how tumor control rate was computed.

To clarify this point, we added the following sentence in the section "Methods/Contrast enhancing tumor progression assessment on MRI":

“As a metric to evaluate tumor control during the treatment period, the slope of enhancing tumor volume was calculated for each patient from the first sonication to the end of inclusion, and expressed in mL/month.”

(18) There is concern that tumor control rate is impacted by the amount of residual tumor after resection. Were the groups balanced on this? It appears as though there might have been differences in baseline mean/median values of hyperintense T1w in implant ROI between the cohorts.

There was no significant difference between mean or median hyperintense volume at baseline between cohorts C and D:

Group “Carbo C”: mean=6.63(\pm 4.81) mL ; median=4.67(iqr:2.94-9.24) mL
Group “Carbo D”: mean=5.01(\pm 6.05) mL ; median=2.56(iqr:1.47-7.04) mL
T-test: p=0.453; Wilcoxon-Mann-Whitney: p=0.090

Although there was no significant difference, as the reviewer noticed, the median volume is slightly lower in cohort D than in cohort C.

(19) Rather than using --- when a value cannot be estimated, please indicate "not estimable".

Data with a longer followup was included in the revised article, so we have no longer used "---".

(20) Please include 95% confidence intervals in Table 3.

Done.

(21) Cannot say that the median OS compares favorably to historical recurrent GBM trial/studies because this is a very select patient population compared to most other recurrent GBM studies. Patients with resectable recurrent tumors as well as recurrent tumors less than 70 mm will have better clinical outcomes than the general recurrent GMB population.

We agree with the reviewer that median OS is typically higher in patients that undergo additional resection surgery. Nevertheless, the data points to a median OS of around 8-12 months for this patient population in recent trials, with trials showing 12 month overall survival typically having a larger number of IDH mutant patients also included. Additional text has been added to the Discussion to address this point.

(22) There is no evidence that cohort D has better clinical outcomes compared to cohort C. By chance, one cohorts will have better values than another. However, the p-values do not indicate significant differences AND the confidence intervals completely overlap for all the clinical outcomes. At best, the findings might support an hypothesis that needs to be tested but there is no evidence that one cohort is better than the other in terms of clinical outcomes. The manuscript needs to be revised to remove misleading comments/implications that cohort D has better clinical outcomes than cohort C. Only place it might be better is in the BBBB effectiveness (which is not a clinical outcome) and perhaps tumor control, which is not measured with a validated measurement.

We agree with the reviewers that the proposed claims can be further tempered by the lack of significance in OS observed between cohorts. We've made significant changes to the Discussion in light of the reviewer's comments.

Reviewers' Comments:

Reviewer #1:

Remarks to the Author:

The authors have clarified the acoustic pressure, electrical power relationship at the individual emitter level. What is still unclear with the 9-emitter device is how the system manages the overlap of the emitter-produced 'cylinders of acoustic pressure', as shown in Figure 1 and 3. Is it assumed and tolerated that certain brain regions may receive 2-3X the amount of mechanical energy? If so, this should be discussed, as well as how this may impact drug delivery. Figure 5 shows a drawn green shape; what this depicts is not described in the figure legend. If the green shape is the representation of the region of acoustic effects, what determines the depth (deep green line) of the acoustic effect from the 'cylinders of acoustic pressure'? If this is not known, then the deep green line should not end sharply, as depicted.

The concluding statement "Our approach has shown the potential to increase local concentrations of therapeutic agents in the brain where the BBB is otherwise intact..." is not supported by this work.

What evidence is provided for increase in local concentration of therapeutic agents, specifically carboplatin, in the brain?. Only evidence of gadolinium contrast agents increasing in the brain is given.

Reviewer #2:

Remarks to the Author:

Thank you for the authors for addressing my remarks and concerns. I believe this manuscript provides noteworthy results and will advance the field of drug delivery and treatment of glioblastoma. The work supports the claims and conclusion and the methodology is sound.

Reviewer #3:

Remarks to the Author:

I have no further question after reviewing the response from the authors.

Reviewer #5:

Remarks to the Author:

The authors have addressed almost all the comments from Reviewer #4. Nevertheless, some minor suggestions and additional comments are raised as below.

Suggestions:

(1) One of the main aims of this study was to identify the MTD defined as the highest active-beams level at which ≤ 1 DLT occurred in a maximum of 6 patients by cohort. However, the study did not find the MTD since no DLTs were observed. Authors should claim clearly about the motivation of the maximum candidate dose settings and why the exact MTD does not matter.

(2) As the comment from Reviewer #4, the basis of using 0.30 as an historical control value seems a little bit arbitrarily. Authors should add this concern (based on historical data/paper, or the consensus from clinicians?) in the part of statistical analysis.

(3) Is there a typo of "cohort D (sonication immediately preceding carboplatin)" in the Patient Profiles? The sonication was performed immediately after the completion of the carboplatin infusion, not preceding carboplatin.

(4) As shown in Figure 6B, there is a significant outlier in cohort C. Although non-parametric tests are not sensitive to the outlier, can this be explained from a clinical perspective?

(5) The upper limit of 95% CI of mOS in the discussion should be 17.3, not ---- (not estimable).

November 8, 2023

Author Response in Blue

Manuscript Title: A phase I/II study of blood-brain barrier disruption in recurrent glioblastoma patients undergoing carboplatin chemotherapy

Corresponding Author: Alexandre Carpentier (alexandre.carpentier@aphp.fr)

Reviewer #1 (Remarks to the Author):

The authors have clarified the acoustic pressure, electrical power relationship at the individual emitter level.

What is still unclear with the 9-emitter device is how the system manages the overlap of the emitter-produced 'cylinders of acoustic pressure', as shown in Figure 1 and 3. Is it assumed and tolerated that certain brain regions may receive 2-3X the amount of mechanical energy? If so, this should be discussed, as well as how this may impact drug delivery.

Although there may be some overlap of beams in the far-field at >5 cm from the transducer, the amount of mechanical energy delivered in these regions is always less than the maximum for a single emitter (at the region of peak acoustic pressure, 15 mm from the implant surface). This is furthermore described below:

- The SonoCloud-9 implant follows the curvature of the skull, and beams from several emitters may overlap, but the nine emitters of the implant are activated sequentially and are never all on at the same time. Due to the limited curvature of the skull, the beam can overlap only in the far-field of the emitters, at about 5 cm from the transducer/inner skull surface, where the acoustic pressure is decreased by a factor of 2 compared to the maximum acoustic pressure. The acoustic pressure in the overlap regions is thus expected to be <0.5 MPa (1.03 MPa is the maximum acoustic pressure, at about 15 mm of the implant, in a region where there is never overlap).
- Furthermore, an analysis performed on the 27 patients of this study with the 9 emitters activated showed that the maximum number of beams overlapping was 3.2+/-0.9. The zone where at least two beams overlapped represented only 9.1+/-3.9% of the total sonicated volume. As expected, a 0.5+/-0.1 MPa maximum acoustic pressure in the zone with overlap was estimated in simulations. The estimated maximum total mechanical energy in the overlap zone, which is proportional to the square of the acoustic pressure, was 0.4+/-0.2 relative to the maximum in the targeted tissues.
- Lastly, we compared post-ultrasound gadolinium enhancement inside and outside of overlap regions for all treatments in this study and found no significant differences. Assuming that gadolinium enhancement is a surrogate for carboplatin delivery (unpublished preclinical results from our team), we thus don't expect significant differences in drug delivery in these regions.

Figure 5 shows a drawn green shape; what this depicts is not described in the figure legend. If the green shape is the representation of the region of acoustic effects, what determines the depth (deep green line) of the acoustic effect from the 'cylinders of acoustic pressure'? If this is not known, then the deep green line should not end sharply, as depicted.

The green shape in Figure 5 corresponds to the targeted volume, where potential carboplatin enhancement is expected. This volume is defined in the Section "Methods/Contrast enhancing tumor progression assessment on MRI": nine 20x80mm cylinders in front of the emitters, which corresponded to the sonicated volume (nine 10x75 mm cylinders) with an additional diffusion margin of 5 mm determined from a preclinical study (reference 44). The definition of the green region will be added in the corresponding figure legend.

The concluding statement "Our approach has shown the potential to increase local concentrations of therapeutic agents in the brain where the BBB is otherwise intact..." is not supported by this work.

What evidence is provided for increase in local concentration of therapeutic agents, specifically carboplatin, in the brain? Only evidence of gadolinium contrast agents increasing in the brain is given.

Indeed, we have shown this in a recent article (Sonabend et al. 2023, Lancet Oncology) using data from patients in this clinical trial, but these results were not explicitly included in the present manuscript. We will revise the last paragraph to the following:

"Our approach has shown the potential to temporarily disrupt the blood-brain barrier using an implantable ultrasound system, thus offering new opportunities for enhancing drug delivery and treating brain diseases with high unmet needs such as GBM. Its clinical efficacy is being further evaluated in a larger pivotal trial (NCT05902169)."

Reviewer #2 (Remarks to the Author):

Thank you for the authors for addressing my remarks and concerns. I believe this manuscript provides noteworthy results and will advance the field of drug delivery and treatment of glioblastoma. The work supports the claims and conclusion and the methodology is sound.

Thank you for taking the time to carefully review our manuscript.

Reviewer #3 (Remarks to the Author):

I have no further question after reviewing the response from the authors.

Thank you for taking the time to carefully review our manuscript.

Reviewer #5 – Replacement for Reviewer #4 – Biostatistics, clinical trials (Remarks to the Author):

The authors have addressed almost all the comments from Reviewer #4. Nevertheless, some minor suggestions and additional comments are raised as below.

Suggestions:

(1) One of the main aims of this study was to identify the MTD defined as the highest active-beams level at which ≤ 1 DLT occurred in a maximum of 6 patients by cohort. However, the study did not find the MTD since no DLTs were observed. Authors should claim clearly about the motivation of the maximum candidate dose settings and why the exact MTD does not matter.

The Maximum Tolerated Dose was defined by number of activated emitters (3 activated emitters out of 9, 6 activated emitters out of 9 or all activated emitters. We confirmed the absence of DLT with all activated emitters (i.e. with the maximum number of emitters present on the device) in 3+3 patients. This is the reason why the exact MTD was not sought.

(2) As the comment from Reviewer #4, the basis of using 0.30 as an historical control value seems a little bit arbitrarily. Authors should add this concern (based on historical data/paper, or the consensus from clinicians?) in the part of statistical analysis.

Indeed, using 0.30 as an historical control value was chosen arbitrarily, in absence of any similar approach in the literature. This threshold however made sense considering the targeted disease.

The manuscript will be modified as follows to clarify this point:

"The proportion (π) of ultrasound sessions that were classified as being successful in opening the BBB was compared to an arbitrary objective performance criterion"

(3) Is there a typo of “cohort D (sonication immediately preceding carboplatin)” in the Patient Profiles? The sonication was performed immediately after the completion of the carboplatin infusion, not preceding carboplatin.

Indeed, this is a typo in the manuscript and has been revised to:

"sonication immediately following carboplatin"

(4) As shown in Figure 6B, there is a significant outlier in cohort C. Although non-parametric tests are not sensitive to the outlier, can this be explained from a clinical perspective?

The patient was a female, 57 years old, IDH wild-type, MGMT unmethylated but with good KPS (90). The time from initial diagnosis was short (6.4 months). The volume of tumor enhancement at cycle one (carboplatin+SonoCloud-9 activation) was quite large as the patient only received a partial resection. None of these patient characteristics could be significant to explain the observed tumor growth rate. The patient was discontinued from the study due based on radiological staging, thus pseudoprogression cannot be excluded.

(5) The upper limit of 95% CI of mOS in the discussion should be 17.3, not ---- (not estimable).

The authors thank the reviewer for pointing out this error; it will be corrected in the manuscript.